# ToMPO: Training LLM Strategic Decision Making from a Multi-Agent Perspective

## Abstract

Large Language Models (LLMs) have been used to make decisions in complex scenarios, where they need models to think deeply, reason logically, and decide wisely. Many existing studies focus solely on multi-round conversations in social tasks or simulated environments, neglecting the various types of decisions and their interdependence. Current reinforcement learning methods struggle to consider the strategies of others during training. To address these issues, we first define a strategic decision-making problem that includes two types of decisions and their temporal dependencies. Furthermore, we propose **T**heory **o**f **M**ind **P**olicy **O**ptimization **(ToMPO)** algorithm to optimize the perception of other individual strategies and the game situation trends. Compared to the Group Relative Policy Optimization (GRPO) algorithm, ToMPO enhances the LLM's strategic decision-making mainly by: 1) generating rollouts based on reasoning the strategies of other individuals, 2) estimating advantages at both the graph-level and sample-level, and 3) balancing global and partial rewards. The ToMPO algorithm outperforms the GRPO method by 35% in terms of model output compliance and cooperative outcomes. Additionally, when compared to models with parameter sizes 100 times larger, it shows an 18% improvement. This demonstrates the effectiveness of the ToMPO algorithm in enhancing the model's strategic decision-making capabilities.

## 1 Introduction

Large Language Models (LLMs) utilize natural language understanding and generation capabilities to achieve leading performance in decision-making scenarios, assisting people in generating (Gou et al., 2024), simulating (Mao et al., 2025), and predicting (Zhang et al., 2024a) decisions across various categories. While LLMs excel in coding and math tasks, they struggle with strategic decision-making, which requires understanding others' intentions, predicting behaviors, and adjusting their own strategies dynamically (Zhang et al., 2024b).

LLMs demonstrate varying strategic abilities in matrix games (Lorè & Heydari, 2024; Herr et al., 2024) and can be enhanced through a game-theoretic workflow (Hua et al., 2024). Recent research further explores LLM strategic decision-making through multi-level thinking (Zhang et al., 2024c; Gou et al., 2024), Theory of Mind (Duan et al., 2024; Cross et al.), task-solving (Zhang et al., 2025a; Wang et al., 2024), as well as influences between individuals and groups (Mi et al., 2025; Zhang et al., 2025b). (detailed related work in section D) These studies provide methods for LLMs to adapt to human society, emerge human behaviors, and serve social issues. However, these studies restrict the strategic decision-making capabilities of LLM to two-agent chatroom environments or single-game scenarios. This approach fails to provide the necessary methods for LLM to enhance its performance in diverse, long-term multi-agent decision-making tasks.

By focusing on these key issues, our paper analyzes the strategic decision-making capabilities of LLMs in complex social environments, where LLMs must sequentially make decisions that impact both individuals and groups. During this period, the prior decisions made by LLM will have a certain degree of influence on subsequent decisions. This implies that individual behaviors may lead to changes in the social structure of the group, and at the same time, changes in the group structure will affect subsequent individual decisions. For instance, in real life, before signing a cooperation agreement with multiple distributors, enterprises will conduct various evaluations. After the coop-

eration agreement is signed, they will implement the cooperation with varying levels of investment over a specified period. Each cooperation has a certain impact on whether the enterprise decides to continue the next collaboration. Furthermore, if there is a desire to terminate the cooperation during the process, it cannot be done immediately; that is, reversing the decision is not possible. This decision-making process helps highlight the real-world challenges faced by individuals and groups over time, posing a challenge to the model's capabilities.

In this context, we first define the problem as a sequential decision-making process that primarily involves graph-level and effort-level decisions. Then we build three kinds of complex social environments to test SOTA (State-of-the-Art) LLM performance. To optimize performance, we propose a reinforcement learning algorithm that integrates a multi-agent perspective into the LLM-based policy model training process. Based on the preliminary tests, we created an expert dataset containing the effort-level decisions made by models that achieve high rewards, across various topological positions and at different stages in the game. The policy model effectively learns decision-making at the effort level from the expert dataset through a supervised fine-tuning process. We enhanced the policy model for graph-level decision-making through reinforcement fine-tuning, which incorporates multi-agent considerations in reward modeling during the training process.

Our contribution can be summarized as:

- We define a problem for real-world strategic decision-making and design corresponding general simulation environments for decision data generation and examination.

- We evaluate the performance of the State-of-the-Art (SOTA) models and provide a dataset including the expert model's strategic decisions under different topological structures and at different game time processes.

- We propose a reinforcement learning algorithm, Theory of Mind Policy Optimization (ToMPO), and apply it to the `Qwen-2.5-7B-instruct` model, achieving improvement in strategic decision-making capabilities.

## 2 PROBLEM FORMULATION

In contrast to the scenarios discussed in Theory of Mind (Strachan et al., 2024; Liu et al., 2025c) and single LLM long-term planning (Huang et al., 2024; Ma et al., 2025), we require the LLM to operate as an agent within a multi-agent environment consisting of at least three agents, making two types of decisions sequentially. During any decision-making process, an agent considers the strategies of other agents and its subsequent strategy, depending on its own state. These considerations will autonomously change based on the agent's social status, game progress, and others' performance.

**Graph-Effort Strategic Decision-Making**   We define the decision-making process as a set $\langle \mathcal{N}, \mathcal{S}, \mathcal{A}, T, \tau, f, r, \gamma \rangle$, with the set of all agents or players $\mathcal{N} = \{1, 2, \ldots, N\}$, state space $\mathcal{S}$, total game round $T$, decision type sequence $\tau$, the state transition function $f \in \{f_G, f_E\}$, utility function $r$, and discount factor $\gamma$. The agent state at round $t$ includes the agent's social relationship structure ($\mathcal{G}$) and effort ($\mathcal{E}$) at this round. $L$ and $M$ represent how many steps of actions related to structure forming and effort investment, respectively in one round. $\tau$ represents the overall action type sequence. For example, when $L = 1, M = 2$, let $\tau = \{(G, E, E), (G, E, E), \ldots\}$ represent a sequence where the LLM agent must make a graph-level decision at step 0 in one game round. This decision involves choosing whether to join one group or establish relationships with others. At steps 1 and 2, the agent will determine how much to invest based on the social relations established in step 0. This pattern continues in subsequent rounds. $a_{i,t,j}$ is the action decision agent $i$ made at step $j$ of round $t$ (equation 2). Action space $\mathcal{A} = \{A_i\}_{i \in \mathcal{N}} = \{a_{i,\tau}\}_{i \in \mathcal{N}}$ (equation 3).

$$\forall i \in \mathcal{N}, \ t \in [0, T-1], \ j \in [0, L+M-1] \tag{1}$$

$$S_t = (G_t, E_t), \quad \tau(t, j) \in \{G, E\}, \quad a_{i,t,j} = \begin{cases} g_{i,t,j} & \text{if } \tau(t,j) = G \\ e_{i,t,j} & \text{if } \tau(t,j) = E \end{cases} \tag{2}$$

$$A_t = (A_G^{t,L}, A_E^{t,M}) = (\{A_G^{t,0}, A_G^{t,1}, ..., A_G^{t,L-1}\}, \{A_E^{t,L}, A_E^{t,L+1}, ..., A_E^{t,L+M-1}\}) \tag{3}$$

**Decision-Making Optimization with Credit Assignment** According to Credit Assignment (Nguyen et al., 2018; Pignatelli et al., 2024) in reinforcement learning, we decompose strategic decision-making into dual complementary processes (equation 4 and 5). $V^*$ represents the optimal value function, and $\mathcal{H}_t = \{(a_\tau, r_\tau)\}_{\tau=0}^{t-1}$ denotes the decision-reward history. The forward process is designed to optimize the model's decision-making capabilities within a defined social graph structure. It effectively involves understanding the rules, accurately predicting or assessing the decisions of other agents, and clearly defining its own strategy. On the other hand, the inverse process significantly enhances the model's ability to determine which group structure it will join next, relying on its memory of past decisions. These two processes align with the credit assignment principle (equation 6).

$$\text{Forward Process (Effort Decision Optimization):} \quad \max_{e_t} \mathbb{E}\left[\sum_{k=t}^{T} \gamma^{k-t} r_k \mid G_t = g\right] \quad (4)$$

$$\text{Inverse Process (Graph Decision Optimization):} \quad \max_{g_t} \mathbb{E}\left[V^*(S_{t+1}) \mid \mathcal{H}_t\right] \quad (5)$$

$$\nabla_\theta J(\theta) = \underbrace{\sum_{t:\tau(t)=\mathcal{E}} \psi_E(\delta_t)}_{\text{Forward Process Credit}} + \underbrace{\sum_{t:\tau(t)=\mathcal{G}} \psi_G(\delta_t)}_{\text{Inverse Process Credit}} + \zeta(\Delta\mathcal{C}) \quad (6)$$

## 3 PRELIMINARY TESTING LLM STRATEGIC DECISION-MAKING

### 3.1 GRAPH-EFFORT STRATEGIC GAME DESIGN

We present two sequential multi-agent game environments where Large Language Model (LLM) agents make decisions over $T$ rounds. Both environments involve $N$ agents making choices related to social graph formation (G) and effort investment (E) to maximize their individual utility. Agents observe full historical information ($G_{history}, x_{history}, \pi_{history}$) to inform their current decisions. The decision-making process in each round generally consists of two key components: graph formation and effort investment. An agent refers to an individual who participates in the game and is part of the graph. As shown in Figure 1, the agent, represented by the policy model, makes decisions simultaneously as other agents in the environment.

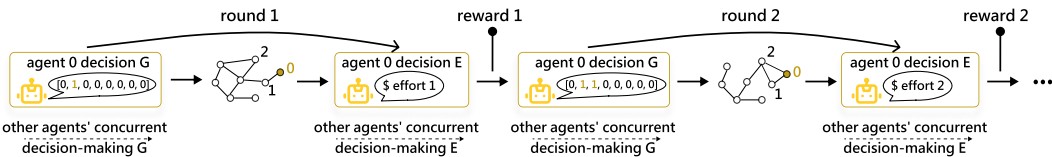

Figure 1: Demonstration of a two-round decision-making process in the GE sub-environment.

### 3.1.1 SEQUENTIAL *BCZ* GAME

This environment extends the classic *BCZ* (Bala-Goyal-Jackson) game (Ballester et al., 2006) to a sequential framework. Each agent $i$ simultaneously decides on their social links and effort investments. The sequence in which these decisions are made defines three sub-environments: GE, GEE, and GGE (detailed in Appendix C.1).

**Decision Components**

- **Link Decision(G):** All agents simultaneously decide on mutual social links, represented by an adjacency matrix $G \in \{0, 1\}^{N \times N}$, where $G_{ij} = 1$ denotes a mutual link between agents $i$ and $j$.
- **Effort Investment(E):** Each agent $i$ invests an effort $x_i \geq 0$.

**Utility Function** As for GE, the utility (payoff) for agent $i$ at a given round, $\pi_i$, is defined as:

$$\pi_i = \alpha_i x_i - \frac{1}{2}x_i^2 + \delta \sum_{j \neq i} G_{ij} x_i x_j - c \sum_{j \neq i} G_{ij}, \quad i, j \in \mathcal{N} \quad (7)$$

where: $\alpha_i > 0$: agent $i$'s individual productivity parameter, $x_i$: effort invested by agent $i$, $\delta > 0$: synergy parameter, representing benefit from interactions, $G_{ij}$: indicates a mutual link between agent $i$ and agent $j$, $c > 0$: cost of maintaining a link.

### 3.1.2 SEQUENTIAL PUBLIC GOODS GAME (PGG)

We implement a sequential LLM-based multi-agent Public Goods Game environment, inspired by classical PGG models (Ledyard et al., 1994; Fehr & Gächter, 2000), incorporating endogenous group formation. Further details are available in the associated code implementation.

**Decision Components**

• **Group Formation (G)** All agents simultaneously decide their preferred group memberships. Agent $i$ submits a binary vector $g_i \in \{0,1\}^N$, where $g_{ij} = 1$ signifies a desire to form a group with agent $j$. A mutual link forms if $g_{ij} = 1$ and $g_{ji} = 1$. Non-overlapping groups $G_t$ are then formed by identifying maximal cliques in the resulting graph; agents not in larger cliques form singleton groups.

• **Effort Investment (E)** Within their established groups $G_{t,k}$, each agent $i$ decides on a continuous effort contribution $x_i \in [0,1]$ into their group's public good.

**Payoff Calculation** The payoff (utility) for agent $i$ in group $G_{t,k}$ at round $t$, $\pi_{i,t}$, is calculated as:

$$\pi_{i,t} = \left( r \cdot \sum_{j \in G_{t,k}} x_{j,t} \right) \Big/ |G_{t,k}| - x_{i,t} \tag{8}$$

where: $r > 1$: public good multiplication factor (e.g., $r = 1.5$), $x_{j,t}$: effort contributed by agent $j$ in round $t$, $|G_{t,k}|$: number of agents in group $G_{t,k}$.

### 3.2 EVALUATION METRICS DEFINITION

To assess the performance of LLM agents in both the *BCZ* and Public Goods Game (PGG) environments, we define three key evaluation metrics: $U_1$ (Compliance), $U_2$ (Strategic Efficiency), and $U_3$ (Cooperative Outcome). These metrics are calculated based on the agents' behavior and the resulting game states over $T$ rounds.

$U_1$**: Compliance (Adherence to Game Rules)** $U_1$ measures how well agents' decisions follow the structural and operational rules of the game. For instance, it penalizes non-zero diagonal entries in the link matrix $G$, which represent self-loops that are not allowed in social graph formation. Additionally, it evaluates the presence of general errors or malformed decisions in the log files. A higher value of $U_1$ indicates a better understanding and execution of the game's mechanics.

$$U_1 = \max\left(1 - \frac{\text{Total Rule Violations}}{\text{Total Possible Checks}}, 0\right) \tag{9}$$

$U_2$**: Strategic Efficiency (Proximity to Individual Optimum)** $U_2$ evaluates how well agents make strategic decisions based on the observed graph structure. It measures the difference between agents' actual effort investments, $x_{\text{actual}}$, and their optimal effort levels, $x^*$, which are determined using optimization methods for *BCZ* and the formula $x^* = \max(0, 1 - |G_{t,k}|/r)$ for *PGG*. The optimal effort is calculated based on the final group structure $G$ in each game. A higher $U_2$ indicates that agents are making rational decisions.

$$U_2 = \max\left(1 - \frac{\|\text{Actual Efforts} - \text{Optimal Efforts}\|_2}{\|\text{Optimal Efforts}\|_2}, 0\right) \tag{10}$$

$U_3$**: Cooperative Outcome (Global Welfare Achieved)** $U_3$ assesses the overall collective performance of the LLM agents by comparing the total payoff achieved in the final round to the maximum theoretically possible total payoff (global optimum) for the respective game. A higher $U_3$ indicates more successful collective action and welfare generation.

$$U_3 = \max\left(\frac{\text{Actual Total Payoff}}{\text{Globally Optimal Total Payoff}}, 0\right) \tag{11}$$

## 3.3 Deficiency for Existing Models

According to the preliminary test result in table 1, we can summarize the deficiency into three points. First, most models cannot generate compliant outputs (U1 test metric). For large models, the limitation is reasoning, while for backbone models, it stems from following the rules. For example, some backbone models generate five numbers in the decision list in a six-agent game. Secondly, when comparing the U2 and U3 metrics (BCZ-2 and PGG), which have an upper limit for the optimal solution, we observe that models perform better in scenarios involving homogeneous agents. In our test logs, the model more easily completes the reasoning chain and generates more comprehensive texts in the BCZ game. Therefore, we use the BCZ game to prepare the expert decision data (details in section 4.1). Thirdly, in comparing the results of BCZ-1 and BCZ-2, the reasoning model can more easily recognize that the current optimal investment has no upper limit. Therefore, a larger effort can be made when the network structure is improved.

Table 1: Large models are tested in three complex social environments, with three simulations each. BCZ-1 optimizes for homogeneous agents without limits, while BCZ-2 suits heterogeneous agents with limits. PGG features isomorphic agents and also has an optimal solution with limits.

| Category | Model Name | U1 | | | U2 | | | U3 | | |
|---|---|---|---|---|---|---|---|---|---|---|
| | | BCZ-1 | BCZ-2 | PGG | BCZ-1 | BCZ-2 | PGG | BCZ-1 | BCZ-2 | PGG |
| LLM | GPT-4o | 0.996 | 0.960 | **1** | 0.254 | 0.845 | 0.660 | 62.831 | 0.007 | 0.445 |
| LLM | DeepSeek-V3 | **1** | **1** | **1** | **0.971** | **0.994** | 0.355 | 18.253 | 0.010 | 0.755 |
| LLM | Llama-3.3-70B | 0.758 | 0.740 | 0.863 | 0.702 | 0.275 | 0.533 | 5.385 | 0.004 | 0.649 |
| LLM | GPT-4o-mini | 0.942 | 0.960 | 0.988 | 0 | 0.014 | **0.672** | 288.208 | 0.007 | 0.554 |
| LRM | GPT-o3 | 0.963 | 0.980 | 0.996 | 0.904 | 0.631 | 0.403 | $2.852 \times 10^9$ | 0.006 | **0.877** |
| LRM | DeepSeek-R1 | 0.996 | 0.980 | 1 | 0.333 | 0.808 | 0.500 | $8.045 \times 10^9$ | **0.033** | 0.750 |
| LRM | kimi-k2-0711-preview | 0.971 | 0.960 | 0.992 | 0.401 | 0.005 | 0.529 | $1.059 \times 10^4$ | 0.001 | 0.531 |
| Backbone | Qwen2.5-7B-instruct | 0.650 | 0.640 | 0.779 | 0.414 | 0.224 | 0.511 | 42.542 | 0.006 | 0.713 |
| Backbone | Llama-3.1-8B | 0.704 | 0.600 | 0.767 | 0.367 | 0.008 | 0.512 | 7.660 | 0.004 | 0.707 |

# 4 ToMPO: Theory of Mind Policy Optimization

## 4.1 Effort Reasoning Learning

Through the preliminary test results, we find that reasoning models are consistently effective at defining the "sub-tasks" necessary to achieve the ultimate goal and complete the overall task. In contrast, backbone models like `Llama-3.1-8B` struggle to reason through a series of steps to finish tasks one by one; they tend to repeat existing rules and perform basic calculations simply. The challenge for the backbone model lies more in transforming the strategic reasoning with social elements into a series of small tasks leading to the final decision, rather than in making the model's calculations more accurate. This is in perfect harmony with the concept of Program of Thought (Chen et al., 2022). The model needs to learn the compliant generation and thinking program first before some other higher needs.

So, according to the model deficiency analysis in section 3.3, we identify the expert models that meet the evaluation criteria U1 and demonstrate a balanced capability in U2 and U3. This means that these models can provide compliant outputs while excelling in both the individual optimal solution and the group optimal solution. We select two reasoning models and analyze their thinking processes to identify a common program of thought for improving reasoning. We organize two programs of thought for decisions regarding graphing and effort, and then we generate expert data using the `GPT-o3` model based on the Program of Thought prompts.

After getting the expert effort decision data ($D_{\text{Effort}}$), we use these data to fine-tune the policy model for learning the common thinking program and compliance output. The optimization method of

Low-rank adaptation (LoRA) fine-tuning (Hu et al., 2022) is shown in Formula 12.

$$\theta^* = \{A^*, B^*\} = \arg\min_{\{A,B\}} \left( -\mathbb{E}_{(x,y) \sim D_{\text{Effort}}} \left[ \sum_{t=1}^{|y|} \log \pi_{(W_0 + \frac{\alpha}{r} BA)}(y_t \mid x, y_{<t}) \right] \right) \quad (12)$$

## 4.2 THEORY OF MIND POLICY OPTIMIZATION (ToMPO)

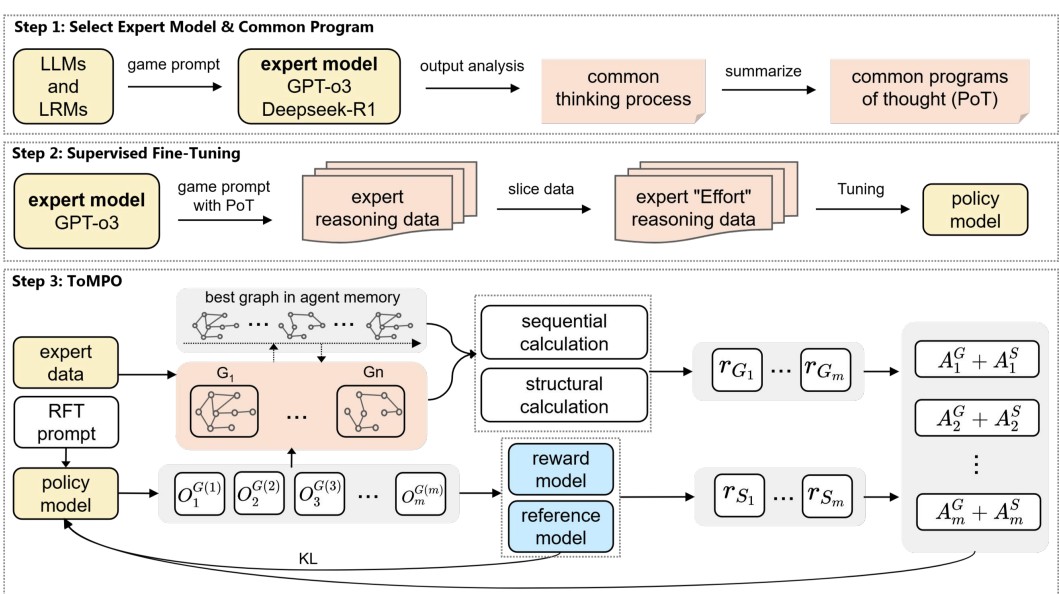

Figure 2: Demonstration of our Theory of Mind Policy Optimization (ToMPO) method. **Step 1:** Select an expert model and common programs of thought. **Step 2:** Supervised Fine-Tuning of the policy model for Effort Decision optimization. **Step 3:** Reinforcement Fine-Tuning policy model with ToMPO algorithm for Graph Decision optimization.

Common policy optimization methods usually calculate advantage from a single agent perspective. This will cause the policy model's adaptability to the environment or information to become increasingly self-centered, to some extent, ignoring the performance and strategies of other models (agents) in the environment. More importantly, when the policy model's decisions involve dependencies among rounds, for example, the decision in round $i + 1$ will be based on the memory of round $i$, the update of the policy model cannot rely solely on the $n$ rollouts of a single round.

As models increasingly resemble human thinking and decision-making, enhancing their capabilities through the Theory of Mind (ToM) (Frith & Frith, 2005; Li et al., 2023; Wu et al., 2025) has garnered significant attention. It's crucial to consider the strategies of other agents during the rollout generation and advantage estimation, as this directly affects the model's policy update process.

**Training Data Preparation:** We consider the policy model (`Qwen-2.5-7B-instruct`) as Agent 0 in all the games during the training process. All other agents are represented by the expert model `GPT-o3`. This makes the strategies of the policy model generally inferior to those of other individuals in the environment, making the purpose of reinforcement learning training clearer. In the model's reinforcement training, classifying the difficulty level of the training data is very important (Pikus et al., 2025). Other agents during the training process directly affect the proportion of the advantages of the policy model's strategy and the learning difficulty. Therefore, we used the expert model to conduct 126 simulations in environments with both homogeneous and heterogeneous agents of different quantities (from 4 to 8), with each simulation lasting for 10 rounds. We collected the actual graph formation situations of each round of the expert models as the "memory" part in the RFT prompt, and the graphs formed by the expert models as the expert data for the reward calculation in the RFT process.

**Concise description for ToMPO algorithm:** Let the generated decision graph by `GPT-o3` models using the same prompt (the same game parameter settings) be the expert data. Then, we have the expert data decision graph $G_{\text{expert}}$, and $m$ rollouts $O$ at step $p$. Each rollout $O$ contains a decision list showing the policy model (Agent 0) strategy. Each list combines with the expert decision graph under this prompt to form a complete graph, denoted as $G_1 \ldots G_n$, representing the final summary of all agents' strategies. At the graph level, each graph $G$ is compared to the $G_{\text{expert}}$ for structural calculations, as well as to the prompt best graph $G_{\text{bprompt}}$ and memory best graph $G_{\text{bmemory}}$ for sequential calculations. At the sample level, each $G$ compares to the $G_{\text{expert}}$ and calculates group advantage. Detailed algorithm process is in Appendix B.

Based on the ToMPO algorithm, the overall optimization objective is:

$$
\mathcal{J}_{\text{ToMPO}}(\theta) = \mathbb{E}_{q \sim \mathcal{P}(Q), \{\boldsymbol{a}_i\}_{i=1}^m \sim \pi_{\theta_{\text{old}}}(\cdot|q)} \left[ \frac{1}{m} \sum_{i=1}^m \min\Big( r_i(\theta), \right.
$$

$$
\left. \text{clip}\big(r_i(\theta), 1-\varepsilon, 1+\varepsilon\big)\Big)\big(w_{\text{S}} A^{\text{S}}(\boldsymbol{a}_i) + w_{\text{G}} A^{\text{G}}(G_i)\big) - \beta \, D_{\text{KL}}[\pi_\theta \| \pi_{\theta_{\text{old}}}] \right]
$$

(13)

ToMPO graph-level advantage estimation balances local precision with global graph optimality, while the sample-level advantage focuses on evaluating the policy model's decisions.

### 4.2.1 REWARDS

Our reward functions contain three parts. We first calculate the Compliance Reward for all rollouts. For those rollouts that are compliant, we calculate the Sample-Level and graph-level rewards.

**Compliance Reward:** We set the basic reward at 0.5 points for model compliance, which means it can generate a decision list where the list length equals the agent sum and there are no self-loops. However, if the model cannot generate the thinking process and the decision list, or if the list does not meet the above needs, the reward is deducted by 1 point, resulting in a final score of -0.5 points.

**Sample-Level Reward:** We believe the sample-level reward needs to be more sensitive to the decision list of the policy model itself. So, we use the F1 score and accuracy to calculate, highlighting the decision list's weight.

$$
R_{\text{sample}}(G) = 5 \left( 0.7 \, F_1(G, \, G_{\text{expert}}) + 0.3 \, \text{Acc}(G, \, G_{\text{expert}}) \right)
$$

(14)

**Graph-Level Reward:** At the graph level, all the comparisons between graphs need to be fair, so we use the Hamming distance for calculation. We calculate and update three rewards, the graph reward $R_{\text{graph}}(G)$, the prompt best reward $R_p^{\text{prompt}}$, the memory best reward $R_p^{\text{memory}}$. $R_{\text{graph}}(G)$ represents the Hamming distance between the actual rollout graph and the expert decision graph. The term $R_p^{\text{prompt}}$ calculates the highest reward among all rollout graphs generated from a single prompt. Meanwhile, $R_p^{\text{memory}}$ is updated whenever a larger reward is obtained within the same game parameter settings (with the exception that only the agent's memory in the prompt is different). $\theta_i$ is the combination of hyperparameters to which the rollout $i$ belongs.

$$
R_{\text{graph}}(G) = 1 - \frac{1}{|\mathcal{E}|} \sum_{(i,j) \in \mathcal{E}} \left| G_{ij} - G_{ij}^{\text{expert}} \right|, \quad |\mathcal{E}| = N(N-1)
$$

(15)

$$
R_p^{\text{prompt}} = \max_{k \in \text{group}(p)} R_{\text{graph}}(G), \qquad \forall p \in \{1, \ldots, M\}
$$

(16)

$$
R_i^{\text{memory}} = \max_{\text{history } \mathcal{H}(\theta_i)} R_{\text{graph}}(G), \qquad \theta_i = (\alpha, \delta, c)_i
$$

(17)

### 4.2.2 ADVANTAGE ESTIMATION

We mainly use the reward at the sample level $R_{\text{sample}}$ to estimate sample-level advantages $A_m^S$, and the reward at the graph level $R_{\text{graph}}$ for graph-level advantages $A_m^G$. In our training, we set the $w_{\text{local}}$, $w_{\text{sample}}$ as 0.8, the $w_{\text{global}}$ and $w_{\text{graph}}$ as 0.2.

$$
A^S(G_i) = \frac{R_{\text{sample}}(G_i) - \text{mean}\big\{R_{\text{sample}}(G_1), \ldots, R_{\text{sample}}(G_n)\big\}}{\text{std}\big\{R_{\text{sample}}(G_1), \ldots, R_{\text{sample}}(G_n)\big\} + \varepsilon}
$$

(18)

$$A^G(G_i) = w_{\text{local}}\Big(R_{\text{graph}}(G_i) - R_i^{\text{prompt}}\Big) + w_{\text{global}}\Big(R_{\text{graph}}(G_i) - R_i^{\text{memory}}\Big) \qquad (19)$$

The overall advantage of a rollout can be calculated as the sum of the sample-level and graph-level advantages, with normalization applied. Compared to the GRPO advantage estimation (Guo et al., 2025; Shao et al., 2024), the ToMPO advantage has two main differences. First, in addition to the sample advantage, we have also improved the graph advantage. This enhances the model's ability to consider the graph more thoroughly while achieving high scores, allowing it to learn more effective decision-making methods. In the rewards at the graph level, we consider both the difference between the current round of the graph and the optimal solution for the same hyperparameters. This allows the model's strategy to gradually move towards both the short-term optimum and the global optimum at the same time.

$$A(G_i) = w_{\text{sample}}A^S(G_i) + w_{\text{graph}}A^G(G_i), \quad w_{\text{sample}} + w_{\text{graph}} = 1 \qquad (20)$$

## 5 EXPERIMENTS

Since the preliminary test revealed that the Qwen model is relatively balanced in terms of performance across all evaluation criteria, we apply the ToMPO algorithm to the `Qwen-2.5-7B-instruct` model, which completes the effort learning fine-tuning process, and compare it with existing models. We conduct each simulation three times, with 20 rounds each, allowing adequate time for model decision-making.

Table 2: Algorithm examination in four environment settings, compared to backbone models, supervised fine-tuning models, and GRPO applied models. **We use the global welfare/ actual simulation rounds to represent BCZ and PGG U3 here, illustrating the efficiency of global welfare gains.**

|  | BCZ - GE | | | BCZ - GEE | | | BCZ - GGE | | | PGG - GE | | |
|---|---|---|---|---|---|---|---|---|---|---|---|---|
|  | U1 | U2 | U3 | U1 | U2 | U3 | U1 | U2 | U3 | U1 | U2 | U3 |
| Deepseek-V3 | 1 | 0.44 | 0.11 | 1 | 0 | 0.09 | 1 | 0 | 0 | 1 | 0 | 0.07 |
| GPT-4o | 1 | 0.36 | 0.10 | 1 | 0 | 0.01 | 1 | 0.07 | -0.11 | 1 | 0 | 0.06 |
| Qwen2.5-72b-instruct | 1 | 0.39 | 0.02 | 1 | 0 | 0.03 | 1 | 0.24 | -0.11 | 0.99 | 0 | 0.07 |
| Qwen3-235b-a22b | 1 | 0.05 | -0.2 | 1 | 0 | 0 | 0.99 | 0 | -0.24 | 0.99 | 1 | 0 |
| Qwen2.5-7B-instruct | 0.65 | 0.38 | 0.08 | 0.95 | 0 | 0.53 | 0.75 | 0 | -0.02 | 0.85 | 0 | 0.10 |
| SFT effort learning | 1 | 0 | -0.09 | 1 | 0 | 0.17 | 1 | 0 | -0.02 | 1 | 0 | 0.10 |
| SFT + GRPO | 1 | 0 | 0 | 1 | 0 | 0.99 | 1 | 0.12 | -0.03 | 1 | 0 | 0.11 |
| **SFT + ToMPO** | **1** | 0 | 0.03 | **1** | 0 | **1.34** | **1** | 0.16 | -0.02 | **1** | 0 | **0.25** |

**Evaluation Environments** Based on our problem definition and environment building, we use the BCZ and PGG games as our examination environments. We create subenvironments by modifying the configuration, which includes variables like **the number of agents**, network hyperparameters such as **private gain sensitivity**, **reciprocity intensity**, **connection costs**, and whether the agents are **homogeneous or heterogeneous**. Our experiment environments set as: BCZ-GE (8 homogeneous agents, alpha = 1, delta = 0.05, c = 0.2), BCZ-GEE (5 heterogeneous agents, alpha = [0.8, 1.8, 1.1, 0.6, 1.5], delta = 0.15, c = 0.4), BCZ-GGE (4 homogeneous agents, alpha = 1, delta = 0.1, c = 0.6), PGG-GE (5 homogeneous agents, r = 1.5).

**Evaluation Models and Algorithms** Based on the preliminary test in table 1, we select models `Deepseek-V3` and `GPT-4o` that have balanced capabilities in the metrics for comparison. Furthermore, we add the `Qwen3-235b-a22b` and `Qwen2.5-72b-instruct` for comparison on the number of parameters and model type. We apply supervised LoRA fine-tuning to the backbone model, the GRPO algorithm to the SFT model, and the ToMPO algorithm to the SFT model. The GRPO algorithm serves as the baseline method, using sample-level rewards as mentioned in section 4.2.1 and sample-level advantage estimation in the GRPO algorithm (Shao et al., 2024).

**Result Analysis** Based on the results in Table 2, we can summarize the performance of the models and algorithms as follows. SFT helps ensure that models generate compliant outputs. The models

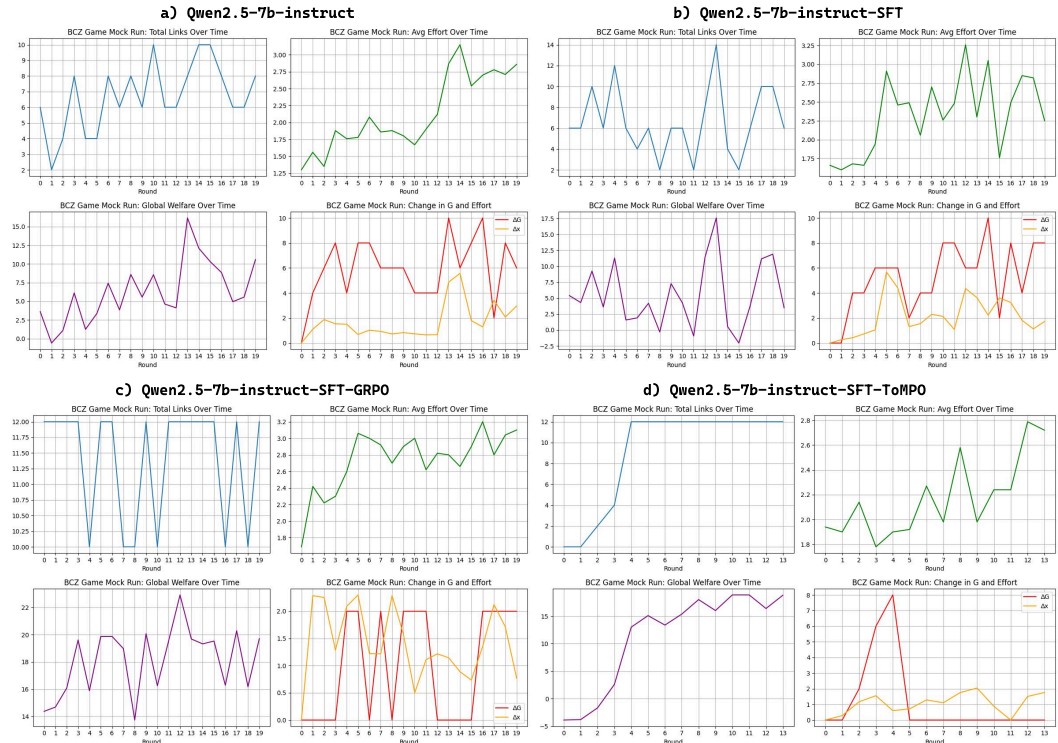

Figure 3: One BCZ-GEE evaluation result comparison for four models: the backbone model (a), the SFT applied model (b), the SFT+GRPO model (c), and the SFT+ToMPO model (d). Each model's results include four components: the blue line shows the total number of links in the graph throughout the game (ending early if unchanged for five rounds), the green line indicates average agent effort, the purple line represents global welfare, and the red and yellow lines display the frequency of changes in the graph and effort, respectively.

generally scored lower on the U2 standard. This is primarily because, in certain scenarios, the models are capable of making higher investments. However, due to the process of mutual exploration and analysis of prior investments made by other models, it becomes challenging for them to make substantial investments directly. As a result, they often deviate from the theoretically optimal individual investment value. Compared to a model with 100 times the parameters, the model trained by ToMPO can achieve the corresponding capabilities.

We analyzed the experimental results and presented the general findings in Figure 3. The result shows the backbone model tends to unpredictable changes in the decision-making process of the graph. It is difficult to make an optimal effort decision under an optimal structure. SFT can help model compliant output, but since the graph does not reach optimality and remains fixed, achieving an average effort that is optimal is challenging. Comparing parts c), a), and b), we find the GRPO algorithm effectively enhances the stability of the model's performance in graph decision-making. On this basis, the model can more easily make the optimal effort decision. When comparing the ToMPO algorithm d) with the GRPO algorithm c), the main takeaway is that the ToMPO algorithm improves the stability and global awareness of the model's decision-making process in graph representation. This enhancement enables the model to make more effective decisions more quickly.

**Limitation and Future Work** Our current work has delivered the supervised fine-tuning (SFT) and ToMPO reinforcement fine-tuning (RFT) on the backbone model, showing the algorithm's effectiveness. The policy model's perspective may be biased towards agent 0 due to our training data. In future work, we will adjust the RFT prompt and training data to broaden the model's perspectives. Despite tests showing reduced capability when combining supervised finetuning for graph and effort, we will explore alternative SFT methods or consider separating the SFT process.

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

## A    LARGE LANGUAGE MODEL UTILIZATION EXPLANATION

In our research, LLM is the backbone and comparison models for the algorithm delivery and examination. We use LLM to generate the configuration, which ensures the balance between randomness and parameter significance. The rest was not involved with LLM.

## B    ToMPO ALGORITHM

---

**Algorithm 1:** Theory of Mind Policy Optimization (ToMPO) Algorithm

---

**Input:** Initial policy model $\pi_\theta$, expert graph $G_{\text{expert}}$, task prompts $Q$, reference model $\pi_\theta^{\text{ref}}$, total training steps $T$, rollout number $m$

1 **for** *iteration* $t = 1, 2, \ldots, T$ **do**
2     Sample prompt $q \sim \mathcal{P}(Q)$;
3     Retrieve expert graph $G_{\text{expert}}$ for prompt $q$;
4     Generate $m$ rollouts: $\{\boldsymbol{a}_i\}_{i=1}^m \sim \pi_{\theta_{\text{old}}}(\cdot|q)$;
5     Construct graphs $\{G_i\}_{i=1}^m$ by combining each $\boldsymbol{a}_i$ with $G_{\text{expert}}$;
6     **for** $i = 1$ **to** $m$ **do**
7         **if** $\boldsymbol{a}_i$ *is compliant* **then**
8             $R_{\text{comp},i} \leftarrow 0.5$;
9             $R_{\text{sample},i} \leftarrow 5\,(0.7\,\text{F1}_i + 0.3\,\text{Acc}_i)$;
10            $R_{\text{graph},i} \leftarrow 1 - \text{Hamming}(G_i, G_{\text{expert}})$;
11            Update $R^{\text{prompt}}$ and $R^{\text{memory}}$ using $R_{\text{graph},i}$;
12            Compute sample-level advantage $A^S(G_i)$ by normalizing $R_{\text{sample},i}$;
13            Compute graph-level advantage $A^G(G_i)$ using $R_{\text{graph},i}$, $R^{\text{prompt}}$, $R^{\text{memory}}$;
14            Combine total advantage $A(G_i) = w_{\text{S}} A^S(G_i) + w_{\text{G}} A^G(G_i)$;
15            Compute importance ratio $r_i(\theta) = \pi_\theta(\boldsymbol{a}_i|q)/\pi_{\theta_{\text{old}}}(\boldsymbol{a}_i|q)$;
16            Update $\theta$ via clip objective with KL penalty $\beta D_{\text{KL}}[\pi_\theta \| \pi_\theta^{\text{ref}}]$;
17        **else**
18            $R_{\text{comp},i} \leftarrow -0.5$;
19            ;

**Output:** Optimized policy model $\pi_\theta^{\text{new}}$

---

## C    ENVIRONMENT AND TRAINING

### C.1    DETAILED DESCRIPTION OF WDBCZ SUB-ENVIRONMENT SEQUENCES

The following are the three sub-environments that define the sequence of decisions made within each round:

1. **GE (Graph-Effort) Environment:** In this environment, each round consists of a single stage of link decisions followed by a single stage of effort decisions.

$$\tau = \{(G, E), (G, E), \ldots\}$$

Agents first decide on their links, forming the graph $G_t$. Subsequently, observing $G_t$, they decide on their effort levels $x_t$.

2. **GEE (Graph-Effort-Effort) Environment:** This environment features a single stage of link decisions, followed by two consecutive stages of effort decisions within each round.

$$\tau = \{(G, E_1, E_2), (G, E_1, E_2), \ldots\}$$

Agents first establish links $G_t$. Then, they make a first effort decision $x_{t,1}$. After all agents have made their first effort decisions (which may be observed by others), they make a second effort decision $x_{t,2}$. The final effort for the round might be a combination of $x_{t,1}$ and $x_{t,2}$ or just $x_{t,2}$ depending on the specific implementation. Our current implementation uses $x_{t,1}$ and $x_{t,2}$ as distinct effort components.

3. **GGE (Graph-Graph-Effort) Environment:** This environment introduces a two-stage linking process, followed by a single stage of effort decisions.

$$\tau = \{(G_P, G_F, E), (G_P, G_F, E), \ldots\}$$

Agents first propose *provisional* links ($G_P$). After observing all provisional link proposals, agents then make *final* link decisions ($G_F$), which forms the actual graph $G_t$. Finally, observing $G_t$, agents decide on their effort levels $x_t$. This allows for a more nuanced negotiation process for link formation.

## C.2 DETAILED TRAINING PARAMETERS

Table 3: Parameters in SFT LoRA training.

| parameter | value |
|---|---|
| lora rank | 64 |
| lora alpha | 32 |
| attention implementation | eager |
| max length | 6000 |
| train batch size | 16 |
| optim learning rate | 5e-5 |

Table 4: Parameters in RFT training.

| parameter | value |
|---|---|
| actor optim learning rate | 1e-6 |
| use kl in reward | true |
| ppo kl coef | 0.1 |
| kl cov ratio | 0.0002 |
| max prompt length | 5500 |
| max response length | 2692 |
| train batch size | 32 |

# D RELATED WORKS

Our work intersects with several active research areas, including the theoretical foundations of credit assignment in reinforcement learning, the burgeoning field of Large Language Models (LLMs) for decision-making, and the complex domain of strategic decision-making in multi-agent systems. This section reviews relevant literature and positions our contributions within these contexts.

## D.1 CREDIT ASSIGNMENT AND POLICY OPTIMIZATION

Credit assignment is a fundamental challenge in reinforcement learning, concerning how to attribute responsibility for outcomes to specific actions or sequences of actions, especially in environments with delayed rewards (Sutton et al., 1998). Early work by Minsky (2007) highlighted this problem, and subsequent research has developed various mechanisms, including eligibility traces (Sutton, 1988) and actor-critic methods (Konda & Tsitsiklis, 1999), to address it. Recently, the concept of credit assignment has been extended to complex, hierarchical, and multi-agent settings (Nguyen et al., 2018; Pignatelli et al., 2024) and Large Language Model agents' social interactions (Yu et al., 2025a). Our work leverages the theoretical underpinnings of credit assignment to decompose the strategic decision-making process into forward (effort decision) and inverse (graph decision) components. This decomposition allows for targeted optimization, where the forward process focuses on immediate utility within a given structure, and the inverse process learns to adapt the structure based on long-term value, aligning with the principles of assigning credit to different types of decisions over time. This approach is distinct from traditional single-agent credit assignment by explicitly considering the interplay between structural and behavioral decisions in a multi-agent context.

Policy optimization methods, such as Proximal Policy Optimization (PPO) (Schulman et al., 2017), MAGRPO (Liu et al., 2025a), multi-conversation DAPO (Yu et al., 2025b), MARFT (Liao et al., 2025), and its variants, have been highly successful in training agents for complex tasks. These methods typically aim to maximize expected cumulative rewards by iteratively updating a policy function. Recent advancements have explored integrating multi-agent considerations into policy optimization, often through centralized training with decentralized execution or by incorporating explicit models of other agents (Lorè & Heydari, 2024). Our Theory of Mind Policy Optimization (ToMPO) algorithm builds upon these ideas by introducing a novel advantage estimation mechanism that explicitly accounts for the strategies and performance of other agents (expert models) in the environment. By incorporating both sample-level (individual decision accuracy) and graph-level (structural optimality) rewards, and by considering historical best performance, ToMPO provides a more nuanced credit assignment mechanism tailored for sequential strategic decision-making in multi-agent social environments, moving beyond standard single-agent or simplified multi-agent PPO formulations.

### D.2 LARGE LANGUAGE MODELS FOR DECISION-MAKING

The remarkable capabilities of Large Language Models (LLMs) in natural language understanding and generation have led to their increasing application in various decision-making scenarios. LLMs have been shown to assist in generating rational decisions (Gou et al., 2024), simulating complex social interactions (Mao et al., 2025), and even predicting outcomes in large-scale social events (Zhang et al., 2024a). Their ability to process and synthesize vast amounts of information, coupled with their emergent reasoning capabilities, makes them powerful tools for augmenting human decision-making or acting as autonomous agents.

However, while LLMs excel in tasks requiring strong logical reasoning (Schmied et al., 2025; Liu et al., 2025b), such as coding and mathematics, their performance in strategic decision-making, particularly in social contexts, remains a significant challenge (Zhang et al., 2024b). This is largely due to the inherent difficulty in understanding others' intentions, predicting their behaviors, and dynamically adjusting one's own strategy in response. Recent efforts have explored enhancing LLM strategic abilities in matrix games (Lorè & Heydari, 2024; Herr et al., 2024) and through game-theoretic workflows (Hua et al., 2024). Furthermore, research has delved into multi-level thinking (Zhang et al., 2024c; Gou et al., 2024), Theory of Mind (ToM) capabilities (Duan et al., 2024; Cross et al.), and task-solving in social environments (Zhang et al., 2025a; Wang et al., 2024). Our work contributes to this growing body of literature by specifically addressing the limitations of LLMs in sequential, long-term multi-agent strategic decision-making, moving beyond two-agent chatroom environments or single-game scenarios. We aim to equip LLMs with the ability to make interdependent decisions that shape and are shaped by evolving social structures, a critical step towards more sophisticated LLM agents in complex social systems.

### D.3 STRATEGIC DECISION-MAKING IN MULTI-AGENT SYSTEMS

Strategic decision-making in multi-agent systems is a rich field (Ma et al., 2024; Yang et al., 2025b; Jin et al., 2025; Liu et al., 2025b; Yang et al., 2025a) that studies how autonomous agents interact and make choices to achieve their objectives, often in the presence of other intelligent agents. Game theory (Hua et al., 2024) provides a foundational framework for analyzing such interactions, offering concepts like Nash equilibrium (Silva, 2024; Holt & Roth, 2004) and Pareto optimality (Censor, 1977) to understand rational behavior (Osborne et al., 2004). Traditional multi-agent reinforcement learning (MARL) has focused on developing algorithms for agents to learn optimal policies in environments where their actions affect others, often dealing with challenges like non-stationarity and credit assignment across agents (Hernandez-Leal et al., 2019).

Recent advancements in MARL have explored more complex social dynamics, including cooperation (Guan et al., 2025), competition (Chen et al., 2024a), operation (Chen et al., 2024b), and the formation of social structures (Yang et al., 2024). Studies have investigated how individual behaviors can lead to emergent group-level phenomena and how group structures, in turn, influence individual decisions (Mi et al., 2025; Zhang et al., 2025b). The concept of Theory of Mind (ToM), which involves an agent's ability to attribute mental states (beliefs, desires, intentions) to others, has gained traction as a crucial component for strategic reasoning in multi-agent settings (Frith & Frith,

2005; Li et al., 2023; Wu et al., 2025). Our research extends these ideas by defining a novel problem of sequential graph-effort strategic decision-making, where agents must make interdependent decisions about both their social connections (graph-level) and their resource investments (effort-level) over time. This problem formulation captures the dynamic interplay between individual actions and evolving social structures, which is often overlooked in simpler multi-agent game settings. By developing ToMPO, we provide a method for LLM agents to learn and adapt their strategies by explicitly considering the actions and potential mental states of other agents, thereby enhancing their ability to navigate and influence complex social environments.

# E  DETAILS FOR PROMPTS

```
You are an autonomous agent participating in a repeated network-effort game simulated by the environment.

Important global rules (read before answering):
- There are two configuration modes controlled by the simulator:
  1) "single" mode: each round has one link decision and then one effort decision.
  2) "multi" mode: each round has one joint link decision and then TWO consecutive effort decisions per agent:
     - Step 1: all agents submit their first-effort e1.
     - After all e1 are submitted, the simulator may publish all agents' e1 values to the environment.
     - Step 2: each agent then submits its second-effort e2, and when answering you may see both your own prev_e1 and the vector prev_e1_all (all agents' first efforts) if the simulator publishes them.

Payoff calculation (per round) you should assume when reasoning:
- single mode (legacy):
    pi_i = alpha[i] * sum_j G[i,j] * effort[j] - c * effort[i]
- multi mode (new):
    pi_i = alpha[i] * sum_j G[i,j] * (effort1[j] + delta * effort2[j]) - c * (effort1[i] + effort2[i])
  where delta is a discount factor (0 ≤ delta ≤ 1) applied to the second effort's benefit to neighbors; costs are paid fully.

Visibility / formatting constraints:
- On step2 in multi mode, you may see prev_e1 (your own first effort) and prev_e1_all (first efforts from all agents) - use them in your reasoning.
- Always output analysis in plain text, and put the requested numeric effort FOR THIS STEP as the very last line of your message in Markdown code format, e.g.:
   `2.5`
- On multi-step runs you will be asked twice; return only the requested effort value for that step at the end of the message.

Safety & parsing:
- Avoid returning any extra Markdown code blocks or formatted numbers except the final single-line numeric code. This helps the simulator parse decisions reliably.

If agents/engineers update reward formulas in code, treat on-code formulas as authoritative; keep system prompt consistent with the implementation above.
```

Figure 4: BCZ game system prompt.

```
You are Agent {agent_id}, in round {round} of the game.

Your task in STEP 1 is to make an initial linking decision (a provisional candidate G_step1).

### Historical Information:
- Group structure history (G_history): {G_history}
- Effort history of all agents: {effort_history}
- Payoff history of all agents: {payoff_history}

You must detail your reasoning about whom you would like to form a mutual link with.

At the end of your message, on a new line, return your provisional linking decision as a binary vector of length N = {num_agents} indicating whom you want to link with.
- 1 means you want to link to that agent.
- 0 means you do not.
- You cannot link to yourself (position {agent_id} must be 0).

Output must be space-separated integers like: `0 1 0 1 0` (no explanation on that line).

Do not swap the order. Always follow [Agent 0, Agent 1, Agent 2, ...] index order.
```

Figure 5: BCZ game graph decision 1 prompt.

```
You are Agent {agent_id}, in round {round} of the game.

Your task in STEP 2 is to finalize the linking decision after observing others' provisional link proposals and
after updates to your memory.

### Observations for STEP 2:
- Everyone's provisional link proposals (G_step1_all): {G_step1_all}
- Your memory has been updated with the provisional links.
- Group structure history (G_history): {G_history}
- Effort history of all agents: {effort_history}
- Payoff history of all agents: {payoff_history}

Please carefully reason and, at the end of your message, on a new line, return your final linking decision
G_final as a binary vector of length N = {num_agents}.
- 1 means you want to link to that agent.
- 0 means you do not.
- You cannot link to yourself (position {agent_id} must be 0).

Output must be space-separated integers like: `0 1 0 1 0` (no explanation on that line).

Do not swap the order. Always follow [Agent 0, Agent 1, Agent 2, ...] index order.
```

Figure 6: BCZ game graph decision 2 prompt.

```
You are Agent {agent_id}, in round {round} of the game.

Current network structure G (this round):
{current_G}

Previous G history:
{G_history}

Previous effort history:
{effort_history}

Previous payoff history:
{payoff_history}

Now decide your FIRST effort for this round to maximize your payoff given the current network and history.

### Output Formatting Requirements
Explain your reasoning in detail. At the **end of your message**, put your chosen first effort (a single
float) on a new line in Markdown code format, e.g.:

`1.5`

Only the final effort number should be in Markdown code format.
```

Figure 7: BCZ game effort decision 1 prompt.

```
You are Agent {agent_id}, in round {round} of the game.

Current network structure G (this round):
{current_G}

Your FIRST effort this round was: `{prev_e1}`

Previous G history:
{G_history}

Previous effort history:
{effort_history}

Previous payoff history:
{payoff_history}

Now decide your SECOND effort for this round to maximize your payoff, taking into account your first effort
above (`prev_e1`). Explain how/why you adjust the second effort relative to your first.

### Output Formatting Requirements
Explain your reasoning in detail. At the **end of your message**, put your chosen second effort (a single
float) on a new line in Markdown code format, e.g.:

`0.8`

Only the final effort number should be in Markdown code format.
```

Figure 8: BCZ game effort decision 2 prompt.

```
You are a rational, goal-oriented intelligent agent participating in a multi-agent Public Goods Game (PGG)
environment with endogenous group formation.

Environment details:

1. There are N agents (numbered from 0 to N-1).
2. Each game consists of T rounds.
3. At the beginning of each round, all agents simultaneously choose their preferred group memberships by
outputting a binary vector of length N:
   - A 1 at position j means the agent wants to form a group with agent j.
   - Groups only form if two agents mutually select each other (both output 1 for each other).
4. The system constructs the final groups by finding maximal cliques (fully connected subgraphs) of mutually
consenting agents, without overlap. Agents not included in any clique form singleton groups.
5. After groups are formed, each agent decides how much effort to contribute to the public good in their
group. The effort is a continuous value between 0 and 1.
6. The payoff for each agent is calculated as:
    payoff_i = (r * sum_{j in group} effort_j) / group_size - effort_i
   where r > 1 is the public good multiplication factor.
7. Historical information about previous rounds' group structures, efforts, and payoffs is provided to you for
decision-making.

Your objective:

- Maximize your individual payoff over the game.
- Make rational and strategic decisions on group formation and effort contribution based on the given
history.

Input format:

- You receive the full history of past rounds: group membership lists, efforts per agent, and payoffs per
agent.
- You know your own agent ID and the total number of agents.

Output requirements:

- For group formation: output a JSON array of length N containing only 0 or 1, representing which agents you
want to group with. Do not include any extra text.
- For effort decision: output a decimal number between 0 and 1 (rounded to two decimals) representing your
effort contribution. Do not include any extra text.

Please adhere strictly to these output formats to ensure correct parsing.

Always act to maximize your expected payoff.
```

Figure 9: PGG game system prompt.

```
[System]
You are an agent participating in a Public Goods Game (PGG). Before each round starts, you need to decide
which agents you want to form a group with.

[Background Rules]
- There are {n_agents} agents in total, numbered from 0 to {n_agents_minus_1}
- You are Agent {agent_id}
- Each agent outputs a binary vector of length {n_agents}
- 1 means you want to group with the corresponding agent
- 0 means you do not want to group with them
- You must set your own position (index {agent_id}) to 0
- A group link forms only if both agents mutually select each other
- The system will find maximal cliques to form the final groups

[Available Historical Information]
{history}

[Your Task]
Decide which agents you want to group with and output ONLY the binary vector.

[Output Format]
1. First, write your reasoning in detail
2. Then, on a new line, write ONLY the binary vector
3. The vector must be in JSON format: a list of {n_agents} integers (0 or 1)
4. The vector must be in agent index order: Agent 0, Agent 1, ..., Agent {n_agents_minus_1}
5. You MUST set position {agent_id} to 0

Example output for Agent 2 in a 5-agent game:
Reasoning: [your detailed reasoning here]
[0, 1, 0, 1, 0]

[Important Instructions]
- Your response must have EXACTLY two parts: reasoning and vector
- The vector must be the last line of your response
- Do NOT include any other text after the vector
- Do NOT include any additional explanations or comments after the vector
- Do NOT include any markdown or formatting in the vector line
- The vector must be valid JSON that can be directly parsed
```

Figure 10: PGG game group decision prompt.

```
[System]
You are an agent participating in a Public Goods Game (PGG). After groups are formed, you need to decide how
much effort to contribute.

[Game Rules]
- You are Agent {agent_id}
- Current group: {current_group}
- Total effort contributed by all group members is multiplied by factor r
- Public return is evenly shared among all group members
- Your payoff = (public return share) - (your effort cost)

[Available Historical Information]
{history}

[Your Task]
Decide your effort contribution and output ONLY the number.

[Output Format]
1. First, write your reasoning in detail
2. Then, on a new line, write ONLY the effort value
3. The effort must be a float between 0.0 and 1.0
4. Round to two decimal places
5. Do not include any units or additional text

Example output:
Reasoning: [your detailed reasoning here]
0.75

[Important Instructions]
- Your response must have EXACTLY two parts: reasoning and number
- The number must be the last line of your response
- Do NOT include any other text after the number
- Do NOT include any additional explanations or comments after the number
- Do NOT include any units like "effort" or "contribution"
- The number must be directly parseable as a float
```

Figure 11: PGG game effort decision prompt.

```
You are participating in a repeated matrix network game as an autonomous agent (LLM).

### Game Setup:
- There are N agents in total, indexed from 0 to N-1.
- In each round, each agent decides:
  1. **Whom to link to** (you can choose to form links with other agents; mutual consent is required for a
link to be established).
  2. **How much effort to exert** (a non-negative real number).

### Payoff Function:
Your payoff πᵢ in each round is calculated as:
  πᵢ = αᵢ * xᵢ - (1/2) * xᵢ² + δ * Σⱼ gᵢⱼ * xᵢ * xⱼ - c * Σⱼ gᵢⱼ

Where:
- xᵢ is your effort in this round.
- gᵢⱼ = 1 if both you and agent j choose to link (mutual), otherwise 0.
- δ > 0 is a complementarity parameter, encouraging cooperation.
- c ≥ 0 is a cost per link.
- αᵢ is your personal linear benefit parameter.

### Objective:
Your goal is to **maximize your earnings(payoff) of the last round**, by trying and reasoning in the previous
rounds:
- What kind of social network (link structure) will benefit you,
- How much effort to contribute given the current structure and history,
- And how other agents might behave.

In each round, the following process occurs:
1. All agents independently decide which other agents to form links with. Links are only formed if both agents
choose to link with each other.
2. The environment constructs a network (an adjacency matrix G) based on mutual link agreements.
3. Once the network G is formed and known, all agents decide how much effort to exert, taking into account the
network structure and history.
4. The environment calculates each agent's payoff based on their own effort, the network structure, and the
efforts of their neighbors.
5. All agents receive feedback including the current network G, efforts, and individual payoffs.

Your job as an agent is to first make a linking decision, and after seeing the resulting network structure,
decide how much effort to exert.
Be strategic. Think long-term. Learn from history.

### Game Parameters:
You have the following information:
- Your personal benefit parameter: **alpha = {alpha}**
- The global complementarity parameter: **delta = {delta}**
- The cost per link: **c = {c}**
Use these parameters in your reasoning and payoff calculation.
```

Figure 12: System prompt for SFT and RFT data generation.

```
You are Agent {agent_id}, in round {round} of the game.

Your task is to decide which agents you would like to form a mutual link with.

### Historical Information:
- Group structure history (G_history): {G_history}
- Effort history of all agents: {effort_history}
- Payoff history of all agents: {payoff_history}

You must detailedly explain your reasoning.

You must follow this thinking program, and based on each program, you can do it by yourself.
# Review the payoff structure:
# Do the Cost-Benefit Analysis:
# Consider the network trend:
# Analysis the agents:
# Consider Pay-off Optimization:

And at the **end of your message**, Please return your final linking decision, which is a binary vector of
length N = {num_agents} **on a new line**, indicating whom you want to link with.
- 1 means you want to link to that agent.
- 0 means you do not.
- You cannot link to yourself (position {agent_id} must be 0).

Output must be space-separated integers like: `0 1 0 1 0` (no explanation in that line).

Do not swap the order. Always follow [Agent 0, Agent 1, Agent 2, ...] index order.
```

Figure 13: Graph decision prompt for SFT and RFT data generation.

```
You are Agent {agent_id}, in round {round} of the game.

The final gain is not required to calculate the cumulative payoff.

Current network structure G (this round):
{current_G}

Previous G history:
{G_history}

Previous effort history:
{effort_history}

Previous payoff history:
{payoff_history}

Based on this, decide how much effort you want to exert this round to maximize your payoff.

### Output Formatting Requirements
You must detailedly explain your reasoning.

You must follow this thinking program, and based on each program, you can do it by yourself.
# Review current network:
# Consider the effort trends and payoff structure:
# Review the payoff function:
# Consider the link costs and risk assessment:
# Analyze the optimal effort:

And at the **end of your message**, place your final effort value **on a new line** at the end of your
response, which a single float number in **Markdown code format**, like:

`2.5`

⚠ Only the final answer should be in Markdown format — use this to help the system identify your chosen
effort. Avoid placing other numbers in the same format elsewhere in your response.
```

Figure 14: Effort decision prompt for SFT and RFT data generation.

