# OpenReview forum: "ToMPO: Training LLM Strategic Decision Making from a Multi-Agent Perspective"
_ICLR.cc/2026/Conference — Submitted to ICLR 2026_

### Official Review · Reviewer_NrgV · 2025-10-22

**Soundness:** 2
**Presentation:** 2
**Contribution:** 3
**Rating:** 2
**Confidence:** 3

**Summary:**

The paper introduces Theory of Mind Policy Optimisation (ToMPO), a reinforcement learning framework for improving LLMs’ strategic decision-making in multi-agent environments. Agents make two interdependent decisions per round, graph-level (who to connect with) and effort-level (how much to invest). The authors extend two games, the BCZ network-effort game and the Public Goods Game (PGG), into sequential multi-agent settings. ToMPO combines:

1. Supervised fine-tuning on expert “Program-of-Thought” data from GPT-o3, and
2. Reinforcement fine-tuning against GPT-o3 agents, optimising both sample- and graph-level advantages.

**Strengths:**

1. Novel framing of multi-agent strategic reasoning: The paper identifies an under-explored LLM capability, reasoning about sequential, interdependent decisions where individual and group incentives co-evolve.
2. Conceptually interesting algorithm: ToMPO attempts to extend PPO/GRPO by introducing Theory-of-Mind style credit assignment, rewarding decisions that align both with self and others’ inferred strategies.
3. Clear hierarchical decomposition: Splitting decision optimisation into forward (effort) and inverse (graph) processes provides an interpretable mapping between short-term utility and long-term social structure.
4. Relevance and timeliness: The work fits within the community’s growing interest in LLMs as agents, multi-agent RL, and emergent social behaviour.

**Weaknesses:**

1. Unclear empirical gain: It is difficult to see where the claimed improvement comes from. In several tables, the ToMPO-trained Qwen model performs similarly or worse than the backbone baseline on key metrics, except for U1 (compliance). The observed increases in U2 or U3 could plausibly result from better formatting or rule adherence rather than genuine strategic improvement. Further ablations on this are required.
2. Ambiguous source of improvement: There are no ablations isolating which components, graph-level advantage, Theory-of-Mind rollouts, or memory rewards, drive the results. Without this, the causal mechanism behind ToMPO’s reported gain is speculative.
3. Lack of clarity and transparency: Important definitions and settings (e.g., BCZ-1/2, “heterogeneous” vs “homogeneous” agents, ToMPO weighting) appear late or only in appendices, making it hard to reconstruct the experiment pipeline.
4. Dependence on GPT-o3: GPT-o3 both supplies the expert data and plays every opponent during reinforcement training. As a result, the learning signal effectively trains Agent-0 (Qwen) to imitate GPT-o3, not to develop independent strategic reasoning.
5. Metrics emphasise compliance, not reasoning: U1–U3 measure formatting validity and numerical optimality, not whether the model actually reasons about others’ incentives or adapts over time. A model could score well without genuine Theory-of-Mind capabilities.
6. Writing and presentation.

**Questions:**

1. BCZ-1 and BCZ-2 are not formally introduced in the main body of text.
2. The difference between BCZ and PGG and what they uniquely offer required further detail.
3. Only upon looking at the prompt did it become clear to what each agent knew and didn't know during the game play. The key information should be included in the main body of text.
4. Were preliminary tests run in self-play or versus GPT-o3 aswell?
5. Did you attempt few-shot prompting before SFT to improve compliance in the weaker models?
6. Are results statistically significant?
7. Its not clear to me exactly how ToMPO captures others’ *mental states* beyond observing their actions. Could you please provide further clarification on this?

---

> ### Author Response · Authors · 2025-11-25
> **Question on BCZ and PGG Game Settings**
>
> Thanks for your professional suggestions and the confirmation on our research motivation. We will answer questions and bring further statistical results.
>
> > Q1. BCZ-1 and BCZ-2 are not formally introduced in the main body of text.
>
> > Q2. The difference between BCZ and PGG and what they uniquely offer required further detail.
>
> We consider two standard variants of the BCZ framework, denoted BCZ-1 and BCZ-2.  BCZ-1 assumes homogeneous agents and yields an unbounded theoretical optimum, whereas BCZ-2 introduces heterogeneous, saturating agents that impose an upper bound on the optimal effort.
>
> BCZ versus PGG in the Graph–Effort Formulation
> In our graph–effort formulation, the key distinction between PGG and BCZ lies in the type of graph action available to agents. A PGG corresponds to a group-choice graph action: the agent selects a participation group, which implicitly induces a fully connected subgraph among group members. By contrast, BCZ corresponds to a link-choice graph action, where the agent explicitly chooses who to cooperate with by selecting edges. These two forms produce different patterns of interdependence: PGG enforces homogeneous, group-level spillovers, whereas BCZ yields dyadic, potentially asymmetric spillovers defined by the chosen links.
>
> This distinction matters because it determines the structure of social externalities that LLM agents must reason about. BCZ allows us to examine whether agents understand localized, partner-specific dependencies, while PGG tests their behavior under uniform, group-based externalities. Together, they cover two fundamental modes of structured group-level cooperation.
>
> We promise to write clearly in the updated version.

---

> ### Author Response · Authors · 2025-11-25
> **Concern on Data Asymmetry among Agents**
>
> > Q3. Only upon looking at the prompt did it become clear what each agent knew and didn't know during the gameplay. The key information should be included in the main body of text.
>
> Unlike typical MAS works [1,2,3], our method decentralizes the training of individual models, allowing them to act as "participants" in choosing their own "position" for the next round, and to determine their own efforts based on the current graph. No single model participating in the game possesses an omniscient perspective or overall control. They are all individuals, relying on past outcome information to make decisions for the next round.
>
> Specifically, each agent knows all the history (network, effort, and payoffs), which only shows the objective outcome and does not include any agent's subjective decision history. This allows agents to make decisions by focusing, analyzing, and summarizing the history, and trying to find the decision behavior of other individuals. We show one example in the RFT dataset for your reference.

---

> ### Author Response · Authors · 2025-11-25
>
> ```
> {"data_source": "network_game", "prompt": [{"role": "user", "content": "You are participating in a repeated matrix network game as an autonomous agent (LLM).\n\n### Game Setup:\n- There are N=6 agents in total, indexed from 0 to N-1.\n- In each round, each agent decides:\n  1. **Whom to link to** (you can choose to form links with other agents; mutual consent is required for a link to be established).\n  2. **How much effort to exert** (a non-negative real number).\n\n### Payoff Function:\nYour payoff πᵢ in each round is calculated as:\n  πᵢ = αᵢ * xᵢ - (1/2) * xᵢ² + δ * Σⱼ gᵢⱼ * xᵢ * xⱼ - c * Σⱼ gᵢⱼ\n\nWhere:\n- xᵢ is your effort in this round.\n- gᵢⱼ = 1 if both you and agent j choose to link (mutual), otherwise 0.\n- δ > 0 is a complementarity parameter, encouraging cooperation.\n- c ≥ 0 is a cost per link.\n- αᵢ is your personal linear benefit parameter.\n\n### Objective:\nYour goal is to **maximize your earnings(payoff) of the last round**, by trying and reasoning in the previous rounds:\n- What kind of social network (link structure) will benefit you,\n- How much effort to contribute given the current structure and history,\n- And how other agents might behave.\n\nIn each round, the following process occurs:\n1. All agents independently decide which other agents to form links with. Links are only formed if both agents choose to link with each other.\n2. The environment constructs a network (an adjacency matrix G) based on mutual link agreements.\n3. Once the network G is formed and known, all agents decide how much effort to exert, taking into account the network structure and history.\n4. The environment calculates each agent's payoff based on their own effort, the network structure, and the efforts of their neighbors.\n5. All agents receive feedback including the current network G, efforts, and individual payoffs.\n\nYour job as an agent is to first make a linking decision, and after seeing the resulting network structure, decide how much effort to exert.\nBe strategic. Think long-term. Learn from history.\n\n### Game Parameters:\nYou have the following information:\n- Your personal benefit parameter: **alpha = 4**\n- The global complementarity parameter: **delta = 0.15**\n- The cost per link: **c = 0.7**\nUse these parameters in your reasoning and payoff calculation.\nYou are Agent 5, in round 2 of the game.\n\nYour task is to decide which agents you would like to form a mutual link with.\n\n### Historical Information:\n- Group structure history (G_history): [Round 0: [[0.0, 0.0, 1.0, 0.0, 0.0, 1.0], [0.0, 0.0, 0.0, 0.0, 0.0, 0.0], [1.0, 0.0, 0.0, 0.0, 0.0, 1.0], [0.0, 0.0, 0.0, 0.0, 0.0, 0.0], [0.0, 0.0, 0.0, 0.0, 0.0, 0.0], [1.0, 0.0, 1.0, 0.0, 0.0, 0.0]], Round 1: [[0.0, 0.0, 0.0, 0.0, 0.0, 1.0], [0.0, 0.0, 0.0, 0.0, 0.0, 0.0], [0.0, 0.0, 0.0, 0.0, 0.0, 0.0], [0.0, 0.0, 0.0, 0.0, 0.0, 0.0], [0.0, 0.0, 0.0, 0.0, 0.0, 1.0], [1.0, 0.0, 0.0, 0.0, 1.0, 0.0]]]\n- Effort history of all agents: [Round 0: [2.86, 0.5, 4.28571429, 0.3, 1.0, 5.5], Round 1: [2.77, 1.0, 3.0, 0.3, 1.79, 4.6]]\n- Payoff history of all agents: [Round 0: [4.42827143, 0.125, 7.6477551, 0.045, 0.5, 11.37021429], Round 1: [2.91485, 0.0, 4.5, 0.045, 0.72305, 9.5664]]\n\nYou must detailedly explain your reasoning.\n\nAnd at the **end of your message**, Please return your final linking decision, which is a binary vector of length N = 6 **on a new line**, indicating whom you want to link with.\n- 1 means you want to link to that agent.\n- 0 means you do not.\n- You cannot link to yourself (position 0 must be 0).\n- Focus, Think and pay attention to which agent you are and your own parameters.\n\nOutput must be space-separated integers like: `0 1 0 1 0` (no explanation in that line).\n\nDo not swap the order. Always follow [Agent 0, Agent 1, Agent 2, ...] index order."}], "ability": "graph_decision", "reward_model": {"style": "custom", "ground_truth": [[0.0, 0.0, 1.0, 0.0, 0.0, 1.0], [0.0, 0.0, 0.0, 0.0, 0.0, 0.0], [1.0, 0.0, 0.0, 0.0, 1.0, 1.0], [0.0, 0.0, 0.0, 0.0, 0.0, 0.0], [0.0, 0.0, 1.0, 0.0, 0.0, 1.0], [1.0, 0.0, 1.0, 0.0, 1.0, 0.0]]}, "extra_info": {"round": 2, "index": 171, "groundtruth": [[0.0, 0.0, 1.0, 0.0, 0.0, 1.0], [0.0, 0.0, 0.0, 0.0, 0.0, 0.0], [1.0, 0.0, 0.0, 0.0, 1.0, 1.0], [0.0, 0.0, 0.0, 0.0, 0.0, 0.0], [0.0, 0.0, 1.0, 0.0, 0.0, 1.0], [1.0, 0.0, 1.0, 0.0, 1.0, 0.0]]}}
> ```

---

> ### Author Response · Authors · 2025-11-25
> **Questions on Algorithm and Preliminary Test**
>
> > Q4. Were preliminary tests run in self-play or versus GPT-o3 aswell?
>
> > Q5. Did you attempt few-shot prompting before SFT to improve compliance in the weaker models?
>
> Preliminary tests run in multi-agent settings, but in every game, all the agents use the same model. For example, if we play the setting as 5 agent, alpha=[1, 0.1, 0.2, 0.2, 0.5], delta=0.05, c=0.2, all five agents use the same model (such as Qwen2.5-7B-instruct), and only the alpha value is different for agents in game setting.
>
> We have tested the few-shot prompting and other guided prompts (such as "When you think the network structure is stable, you can boldly make an investment."). However, models with 7B around parameters reason well but decision bad. For example, model thinking process mentions "now it's a 6 agents game",with a decision array [0,0,0,1] (not match agent number) or mentions "I will link with all other agents" with a decision array [1,0,0,0,1,1] (not all 1 except itself). We think this is the problem mentioned by these work [4,5], and can be solved to some extent by RFT process.
>
>
>
> > Q6. Dependence on GPT-o3: GPT-o3 both supplies the expert data and plays every opponent during reinforcement training. As a result, the learning signal effectively trains Agent-0 (Qwen) to imitate GPT-o3, not to develop independent strategic reasoning.
>
> The algorithm does not rely entirely on the performance of the o3 model because the decision matrix output by the expert model is only used as ground truth to construct a new decision matrix based on the decision array of the policy model. The o3 model is not involved in training and does not generate decision arrays synchronously with the policy model.
>
> Because the o3 model demonstrated better decision-making ability in the preliminary tests and outperformed the R1 model in Program of Thought (PoT) [6], it was chosen as the expert model. Since the ground truth matrix does not involve manually calculating the theoretically optimal solution but rather using the actual results from the model, we will select the model with the better performance.

---

> ### Author Response · Authors · 2025-11-25
>
> > Q7. Its not clear to me exactly how ToMPO captures others’ mental states beyond observing their actions. Could you please provide further clarification on this?
>
> We thank the reviewer for raising this important question. ToMPO goes beyond mere action-tracking by explicitly modeling latent mental-state variables—in particular, the beliefs, desires/goals, and intentions of other agents [7]—rather than only their observable behavior. Concretely, ToMPO learns to infer the behavioral motivations of other agents in observations and updates a structured belief state over others’ internal preferences and future plans, and it uses this belief state recursively during rollouts.
>
> Although the RFT process focuses on Inverse Process (Graph Decision Optimization), the Effort Decision has not been interrupted. Both kinds of decision-making show the BDI (Belief-desire-intention) thinking process [8]. We bring cases for further reference.
>
> Case 1: ToMPO Link Decision
>
> [Agent 0] Link decision:
> ```
> Based on the historical information, it is evident that the agents have generally been choosing to link with a limited number of other agents, with Agent 0 and Agent 6 being the most commonly linked agents. The efforts exerted by the agents have also been decreasing over time, with Agent 0 expending the most effort in the initial rounds.
> Given that Agent 0 has a personal benefit parameter (α₀) of 3 and a cost per link (c) of 0.9, it is crucial for Agent 0 to balance the potential gains from linking with other agents with the costs of doing so. The global complementarity parameter (δ) is 0.12, which encourages cooperation between agents.
> To maximize Agent 0's payoff, it is essential to consider the efforts of other agents and the network structure. Since the efforts have been decreasing over time, it is possible that the agents have reached a point of diminishing returns, where further efforts would no longer significantly increase their payoffs. Therefore, it is advised for Agent 0 to focus on building and maintaining relationships with a few selected agents rather than trying to link with everyone.
> Based on the historical information, Agent 0 has been linked with Agent 3 in Round 0, Agent 2 in Round 1, Agent 3 in Round 2, Agent 5 in Round 6, and Agent 3 in Round 8. Agent 0 has not been linking with anyone else in the past rounds. Given this information, it appears that Agent 3 has been a relatively reliable partner for Agent 0, while the efforts of other agents have not justified the additional cost of linking with them.
> With this in mind, it is recommended for Agent 0 to continue linking with Agent 3 and not to link with any other agents. This decision is based on the following factors:
> 1. Agent 3 has been a reliable partner in the past, suggesting that there is a potential for a mutually beneficial relationship.
> 2. The efforts of other agents have been decreasing, indicating that they are not likely to significantly increase their payoffs with further efforts.
> 3. The cost per link is relatively high, and Agent 0 should be cautious about the number of links it forms.
> Therefore, Agent 0's decision on whom to link with is:
> `0 0 0 1 0 0 0`
> ```
> Case 2: GRPO Effort Decision
>
> [Agent 4] Effort decision:
> ```
> To determine the best effort for this round, I will analyze the current network structure and the previous efforts. Here's a detailed explanation of my reasoning:
>
> 1. **Network Structure Analysis**: The current network structure G shows that there are several links established between the agents. Specifically, agents 1, 2, 3, and 4 have links with agent 5, while agents 1, 2, 3, and 5 have links with agent 4. This suggests that there is a potential for increased payoff by increasing efforts on these links.
>
> 2. **Effort History Analysis**: The previous effort history shows that agents 1, 2, 3, and 5 have been increasing their efforts over the past few rounds. This indicates that they are likely to continue to do so in the next round.
>
> 3. **Payoff Calculation**: The payoff formula for single mode is given by:
>    \[
>    \pi_i = \alpha[i] \times \sum_j G[i,j] \times \text{effort}[j] - c \times \text{effort}[i]
>    \]
>    Since the cost (c) is not provided, we will focus on the term involving efforts. The term \(\sum_j G[i,j] \times \text{effort}[j]\) represents the total effort received from other agents.
>
> 4. **Decision Making**: Based on the analysis, I will focus on increasing efforts on the links that have been established and are likely to continue to increase their efforts. Therefore, I will increase my effort on links with agents 1, 2, 3, and 5.
>
> Taking all of this into account, I will increase my effort to 2.0 in the next round.
>
> Therefore, my chosen first effort for this round is:
> `2.0`
> ```

---

> ### Author Response · Authors · 2025-11-25
> **References**
>
> [1] Liao, J., Wen, M., Wang, J., & Zhang, W. (2025). Marft: Multi-agent reinforcement fine-tuning. *arXiv preprint arXiv:2504.16129*.
>
> [2] Liu, S., Chen, T., Liang, Z., Lyu, X., & Amato, C. (2025). Llm collaboration with multi-agent reinforcement learning. *arXiv preprint arXiv:2508.04652*.
>
> [3] Wan, Z., Li, Y., Wen, X., Song, Y., Wang, H., Yang, L., ... & Wen, Y. (2025). Rema: Learning to meta-think for llms with multi-agent reinforcement learning. *arXiv preprint arXiv:2503.09501*.
>
> [4] Jin, M., Luo, W., Cheng, S., Wang, X., Hua, W., Tang, R., ... & Zhang, Y. (2025, July). Disentangling memory and reasoning ability in large language models. In *Proceedings of the 63rd Annual Meeting of the Association for Computational Linguistics (Volume 1: Long Papers)* (pp. 1681-1701).
>
> [5] Mancoridis, M., Weeks, B., Vafa, K., & Mullainathan, S. (2025). Potemkin Understanding in Large Language Models. *arXiv preprint arXiv:2506.21521*.
>
> [6] Chen, W., Ma, X., Wang, X., & Cohen, W. W. (2022). Program of thoughts prompting: Disentangling computation from reasoning for numerical reasoning tasks. *arXiv preprint arXiv:2211.12588*.
>
> [7] Chen, R., Jiang, W., Qin, C., & Tan, C. (2025). Theory of Mind in Large Language Models: Assessment and Enhancement. *arXiv preprint arXiv:2505.00026*.
>
> [8] ```Belief–desire–intention software model https://en.wikipedia.org/wiki/Belief%E2%80%93desire%E2%80%93intention_software_model```

---

### Official Review · Reviewer_8sLf · 2025-10-23

**Soundness:** 2
**Presentation:** 2
**Contribution:** 2
**Rating:** 4
**Confidence:** 3

**Summary:**

The paper introduces ToMPO, a novel reinforcement learning algorithm for training LLMs in strategic multi-agent decision-making settings. The authors formalize a graph-effort decision process where an agent must first decide on social connections (graph decision) and then make resource allocation or strategy decisions (effort decision). ToMPO leverages a multi-agent "theory of mind" perspective via agent rollouts, dual-level credit assignment (graph- and sample-level advantages), and a custom reward design that balances global and individual outcomes.

**Strengths:**

Introduces a novel class of multi-agent LLM decision problems involving dynamic social graphs and sequential choices.
ToMPO creatively incorporates multi-agent reasoning into policy gradients via dual-level advantage estimation.
A well-designed reward structure that balances local and global goals effectively stabilizes RL fine-tuning.

**Weaknesses:**

The paper lacks confidence intervals or variance estimates. Please report 95% confidence intervals over multiple seeds. Statistical significance is critical for validating the results. I may raise my score substantially if the reported gains are statistically robust across diverse setups.
No ablation studies to disentangle effects of key components (e.g., graph-level advantage, ToM rollouts).
Depends on expert data for SFT; unclear how performance degrades with weaker or fewer demonstrations.
Baseline comparisons are limited.
Evaluation remains in custom environments; unclear generalization to more realistic or diverse strategic tasks.
Dense notation and undefined evaluation metrics slightly hinder accessibility.

**Questions:**

How sensitive is ToMPO’s performance to the quality or quantity of expert demonstrations? Could it work with noisy or imperfect experts?
Can you report ablations for ToMPO’s components (graph-level advantage, rollout reasoning)? Which parts are most critical?
Would your method generalize to agents trained symmetrically (not just Agent 0)? Could one model handle all agents?
How was the 0.8/0.2 reward weighting chosen? Were other hyperparameter combinations tested?

---

> ### Author Response · Authors · 2025-11-25
> **Concern on Expert Model**
>
> We appreciate your detailed suggestions and questions; all of these concerns are important for us to discuss and enrich our article. We answer the three questions first and bring further statistical results. Thanks for your consideration and time.
>
>
> > Q1. How sensitive is ToMPO’s performance to the quality or quantity of expert demonstrations? Could it work with noisy or imperfect experts?
>
> The algorithm does not rely entirely on the performance of the o3 model because the decision matrix output by the expert model is only used as ground truth to construct a new decision matrix based on the decision array of the policy model. The o3 model is not involved in training and does not generate decision arrays synchronously with the policy model.
>
> Because the o3 model demonstrated better decision-making ability in the preliminary tests and outperformed the R1 model in Program of Thought (PoT) [1], it was chosen as the expert model. Since the ground truth matrix does not involve manually calculating the theoretically optimal solution but rather using the actual results from the model, we will select the model with the better performance.
>
> You raised a very crucial question. If it's a noisy expert, the difference in decision-making ability between the policy model and the expert model in the final decision matrix will decrease. Since the decision matrix is based on the intentions of both parties, the poor strategy of the noisy model will affect the training process of the policy model, allowing even poor decisions to receive good rewards during training. However, since the corresponding historical best decision and the round best decision change at the same time, the essence of the algorithm will not change due to the performance of the expert model. In other words, if we use a noisy model's decision as an expert decision matrix (groundtruth), any algorithm will be affected because the calculation of reward is affected.

---

> ### Author Response · Authors · 2025-11-25
> **Concerns on ToMPO Algorithm**
>
> > Q2. Can you report ablations for ToMPO’s components (graph-level advantage, rollout reasoning)? Which parts are most critical? Would your method generalize to agents trained symmetrically (not just Agent 0)? Could one model handle all agents?
>
> Yes, one model can handle multiple roles' perspectives. The ToMPO algorithm is a whole, not composed of modules, so it cannot be broken down into graph-level and sample-level (Only sample-level is GRPO). However, we can gain insights from the training process data of the ToMPO and GRPO algorithms. The algorithm helps the model to **(1) train from multiple perspectives during training, paying attention to the performance of other individuals in the environment from the perspective of any role; (2) consider the historical optimal solution of its own decision, to obtain a better reward earlier and converge to a better result faster.**
>
> Although Theory of Mind is an abstract conceptual content, we incorporate the core of ToM into the model training process with these two points. We explain in detail:
>
> **(1) Multi-role perspective training** is reflected in the training dataset, which includes problems from the perspectives of agents 0 to 7 and six different game settings (different numbers of agents/agent heterogeneity/network connection penalty hyper-parameters). Simultaneously, when calculating the reward, the decision array corresponding to the agent's label is used to replace the decisions of the corresponding expert model agent in the ground truth, forming a new decision matrix G (only the decision part of the numbered agent is generated by the policy model; the rest are decisions of the expert model).
> During training, the performance of other individuals is considered when merging the new decision matrix. Both the policy model and the expert model (which already exists in the ground truth) must agree to cooperate for the decision matrix to show cooperation; otherwise, even if the policy model believes cooperation/non-cooperation is appropriate, unilateral success is not possible. This aligns to some extent with both game theory processes and real-life cooperative substitution scenarios.
>
> **(2) Considering the historical optimal solution of its own decision**, the algorithm will store the policy model under each network hyper-parameter as the optimal solution that the role has reached in the process of advantage estimation. Although this is different from directly calculating the theoretical optimal solution manually, it can be regarded as the "historical optimal solution" of the policy model in solving this multi-agent network problem, in which the policy model is itself a party.
> Although the historical optimal is better than the decision result of a certain round in most cases, it is not ruled out that, due to special circumstances (such as the sudden non-cooperation of other agents in a certain round for no reason), the policy model may not follow the historical optimal decision, but make an adaptive decision. Therefore, after comparing the current round's optimal and the historical optimal, the algorithm does not directly use the historical optimal as the decision for this round, but uses the calculation of the "optimal solution for this round" to reduce the possibility that the historical optimal will affect the agent's adaptive decision.
>
> Our reward calculation involves the Deepseek-v3 model's evaluation of the policy model's thinking process (this is the part within "Compliance Reward"), as well as the o3 model's actual operation as ground truth in the preparation work. Although two different types of models participate in the policy model's update process in a non-trained manner, the training results still show that the ToMPO algorithm achieves a better reward faster and converges to the better result faster than the baseline GRPO algorithm. We currently provide training score data for steps 10, 20, 30, 40, and 50 for observation and promise to add detailed data comparison charts of the training process in the revised version of the paper.
>
> | | | | | | | |
> |:-:|:-:|:-:|:-:|:-:|:-:|:-:|
> ||ToMPO_max|ToMPO_min|ToMPO_mean|GRPO_mean|GRPO_min|GRPO_max|
> |10|5.65|-0.50|3.00|2.51|-0.60|5.65|
> |20|5.65|-0.20|4.25|4.01|-0.20|5.30|
> |30|5.30|-0.20|4.68|4.15|-1.50|5.30|
> |40|5.30|0.80|5.04|4.64|-1.50|6.00|
> |50|5.30|0.80|5.12|4.90|-1.50|5.65|

---

> ### Author Response · Authors · 2025-11-25
> **Question about Hyper-parameters**
>
> > Q3. How was the 0.8/0.2 reward weighting chosen? Were other hyper-parameter combinations tested?
>
> Our experiments revealed that policy models achieve sample-level advantages more easily than graph-level advantages through rollout. This is because sample-level advantages are primarily numerical and confined to the same rollout group, while graph-level advantages involve comparisons with the historical best solutions of the same group and the network's hyper-parameters. For a model to gain an advantage at the graph level, it needs to learn graph-level information. This is also a difficult decision-making challenge when the model itself is an **actor** rather than an observer, making strategic decision-making a very important topic of discussion in multi-agent environments.
>
> Therefore, we aim for the model to dedicate most of its "energy" to learning at the sample level, and then use the remaining energy for graph-level learning, ultimately achieving a stable and appropriate advantage at both levels. This setup facilitates stable model training. We observed that with settings 2-8, the model can complete graph-level learning, allowing us to maintain the hyper-parameter settings without modification. We tested hyper-parameters of 4-6, but this was challenging for the model to learn in the initial stages of training. Results showed significant reward collapse at steps 17, 42, and 47. However, the model trained more stably with hyper-parameters of 2-8.
>
>
> [1] Chen, W., Ma, X., Wang, X., & Cohen, W. W. (2022). Program of thoughts prompting: Disentangling computation from reasoning for numerical reasoning tasks. *arXiv preprint arXiv:2211.12588*.

---

### Official Review · Reviewer_1EjA · 2025-10-28

**Soundness:** 2
**Presentation:** 1
**Contribution:** 3
**Rating:** 4
**Confidence:** 3

**Summary:**

This paper provides a variation of GRPO that factors in ToM for more effective strategic decision-making. It is well-motivated, rigorous, and provides significant improvements in the areas where it is applied (namely, cooperative settings).

**Strengths:**

1. Original contribution with strong numbers and improvements over baselines.
2. Well-founded, rigorous work.
3. Evaluation includes a mixture of open-source and closed-source models.

While these are short sentences, I need to emphasise that this is an original, exciting contribution. These strengths make the paper very robust--provided that the two major weaknesses below are addressed.
In my opinion, employing ToM for reinforcement learning is a good idea that can make good contribution to MAS. What matters from this paper is that there are multiple experimental settings in which ToMPO is shown to work.

**Weaknesses:**

In terms of contributions, this paper is strong. However, it is very poorly written, and the reproducibility of the work is only partial. Both are major--but not impossible to fix--weaknesses.

Writing:
- The flow of the paper needs to be improved: there are a lot of places where things are introduced before/after/referred to without a logical succession flow. For example, the paper starts by framing the problem as (L84): 'In contrast to the scenarios discussed in Theory of Mind': which scenarios? If you have a fully-fledged formal definition of your scenario (L91+) it would make sense to compare both the scenarios and your reframing.
- Overall the paper can be made far more readable by moving the notation to the appendix, and the related works section back to the main body. It defines names that are used but not defined in the main body of the paper, and provides framing for your entire research (see, e.g., my questions from above).
- The main contribution (L72) indicates the introduction/definition of a (new) problem for decision-making. However, the entire work relies on an abstract notion of the data. Given that LLMs are sensitive to natural language, explaining _what_ this problem is (or adding examples of the datapoints) in the main body of the paper would be helpful; and likely will require rephrasing L72.
- The framing takes about eight pages, and the results about one page. There is no discussion of the work, or conclusions drawn. As it stands, it reads more like a technical report, rather than work following the scientific method.
- Do you need the definition of LoRA (L273)?

Minor:
- There are plenty of sentences that may be simplified to improve the flow of the work (e.g., L54, repeat 'cooperation', L63, repeat 'performance').
- Abbreviations are introduced and re-introduced often, including obvious ones like 'Large Language Models'. Sometimes they are introduced backwards (L62; 'State-of-the-art (SOTA)' should be backwards and 'State' is not capitalised)
- Dangling parenthesis in L43
- L110: equation -> equations
- L116: 'these two processes align with the credit alignment principle' (and a referral to an equation). Since CAP refers to an equation, it is better to emphasise it ('the _credit alignment principle_ (eq. X)' or 'the credit alignment principle from eq. X'). This will vastly improve readability since there are plenty of definitions thrown around.
- C.2 tables: maintain consistent capitalisation. Variables need to be explained.
- B: the algorithm needs an explanation in natural language
- Significant digits are inconsistent (see, e.g., Table 2)
These are just examples: the paper needs further, thorough proofreading

_Reproducibility_: while there is pseudocode and prompts, there is no code released even though this is an algorithm that relies on stochasticity. Code release is important in this case, because--without formal proofs of correctness--the only other way to ascertain that this paper does succeed at what it claims is by examining the code, including whether any random seeds were set.
Since this relies on synthetic data generation (which was admittedly skewed, as per the authors), and it is claimed as part of the contributions (L74), this should be released as well. I'm fairly certain that, since the authors make references to the 'associated code implementation', this was solely an oversight on their part.

As I said, the paper's technical contributions are solid. In its current state, it is not of sufficient quality and the lack of reproducibility does put some doubts in this work's trustworthiness, which limits the reviewer's assessment of the work. However, I also believe that the drawbacks could be very easily addressed.

**Questions:**

L468 -- 'compared to a model with 100 times the number of parameters' -- does this mean not the comparison between backbone and finetuned Qwen? Which model is this?

---

> ### Author Response · Authors · 2025-11-25
> **Questions on Model Comparison**
>
> We appreciate the detailed and useful advice on writing and our paper workflow. Thanks for your patience and consideration of our research motivation and paper.
>
> > Q1. L468 -- 'compared to a model with 100 times the number of parameters' -- does this mean not the comparison between backbone and finetuned Qwen? Which model is this?
>
> This is the interpretation of the evaluation data. It is about the comparison between qwen+ToMPO (about 7B) and Deepseek-V3 (about 670B). According to our preliminary test, many small parameter models (about 7B) can not deal with the compliance generation, no mention the graph-level strategic decision-making. So, after we find the training result shows qwen+ToMPO can deal with the compliance generation, we need to compare the small parameter model performance to the large parameter models. If a small model can accomplish tasks that only a model with a large number of parameters can, then we can further summarize and reflect on the relationship between model capability and the number of parameters [1].

---

> ### Author Response · Authors · 2025-11-25
>
> > Q2. Since this relies on synthetic data generation (which was admittedly skewed, as per the authors), and it is claimed as part of the contributions (L74), this should be released as well. I'm fairly certain that, since the authors make references to the 'associated code implementation', this was solely an oversight on their part.
>
> > Q3. As I said, the paper's technical contributions are solid. In its current state, it is not of sufficient quality, and the lack of reproducibility puts some doubts in this work's trustworthiness, which limits the reviewer's assessment of the work. However, I also believe that the drawbacks could be very easily addressed.
>
>
> | | | | | | | |
> |:-:|:-:|:-:|:-:|:-:|:-:|:-:|
> ||ToMPO_max|ToMPO_min|ToMPO_mean|GRPO_mean|GRPO_min|GRPO_max|
> |10|5.65|-0.50|3.00|2.51|-0.60|5.65|
> |20|5.65|-0.20|4.25|4.01|-0.20|5.30|
> |30|5.30|-0.20|4.68|4.15|-1.50|5.30|
> |40|5.30|0.80|5.04|4.64|-1.50|6.00|
> |50|5.30|0.80|5.12|4.90|-1.50|5.65|
>
> We further supplemented the dataset with test questions for multiple individuals, retrained the model, and the results also showed that the ToMPO algorithm reaches a higher reward faster and converges to a better value more quickly than the GRPO algorithm. We will release the decision-making dataset after our updated version article. Thanks for your trust and understanding of our work and research questions.
>
> [1] Xiao, C., Cai, J., Zhao, W., Lin, B., Zeng, G., Zhou, J., ... & Sun, M. (2025). Densing law of llms. *Nature Machine Intelligence*, 1-11.

---

> ### Author Response · Authors · 2025-11-25
> **Explain for Dataset**
>
> We also provide one example of the RFT dataset for further reference.
>
> ```
> {"data_source": "network_game", "prompt": [{"role": "user", "content": "You are participating in a repeated matrix network game as an autonomous agent (LLM).\n\n### Game Setup:\n- There are N=6 agents in total, indexed from 0 to N-1.\n- In each round, each agent decides:\n  1. **Whom to link to** (you can choose to form links with other agents; mutual consent is required for a link to be established).\n  2. **How much effort to exert** (a non-negative real number).\n\n### Payoff Function:\nYour payoff πᵢ in each round is calculated as:\n  πᵢ = αᵢ * xᵢ - (1/2) * xᵢ² + δ * Σⱼ gᵢⱼ * xᵢ * xⱼ - c * Σⱼ gᵢⱼ\n\nWhere:\n- xᵢ is your effort in this round.\n- gᵢⱼ = 1 if both you and agent j choose to link (mutual), otherwise 0.\n- δ > 0 is a complementarity parameter, encouraging cooperation.\n- c ≥ 0 is a cost per link.\n- αᵢ is your personal linear benefit parameter.\n\n### Objective:\nYour goal is to **maximize your earnings(payoff) of the last round**, by trying and reasoning in the previous rounds:\n- What kind of social network (link structure) will benefit you,\n- How much effort to contribute given the current structure and history,\n- And how other agents might behave.\n\nIn each round, the following process occurs:\n1. All agents independently decide which other agents to form links with. Links are only formed if both agents choose to link with each other.\n2. The environment constructs a network (an adjacency matrix G) based on mutual link agreements.\n3. Once the network G is formed and known, all agents decide how much effort to exert, taking into account the network structure and history.\n4. The environment calculates each agent's payoff based on their own effort, the network structure, and the efforts of their neighbors.\n5. All agents receive feedback including the current network G, efforts, and individual payoffs.\n\nYour job as an agent is to first make a linking decision, and after seeing the resulting network structure, decide how much effort to exert.\nBe strategic. Think long-term. Learn from history.\n\n### Game Parameters:\nYou have the following information:\n- Your personal benefit parameter: **alpha = 4**\n- The global complementarity parameter: **delta = 0.15**\n- The cost per link: **c = 0.7**\nUse these parameters in your reasoning and payoff calculation.\nYou are Agent 5, in round 2 of the game.\n\nYour task is to decide which agents you would like to form a mutual link with.\n\n### Historical Information:\n- Group structure history (G_history): [Round 0: [[0.0, 0.0, 1.0, 0.0, 0.0, 1.0], [0.0, 0.0, 0.0, 0.0, 0.0, 0.0], [1.0, 0.0, 0.0, 0.0, 0.0, 1.0], [0.0, 0.0, 0.0, 0.0, 0.0, 0.0], [0.0, 0.0, 0.0, 0.0, 0.0, 0.0], [1.0, 0.0, 1.0, 0.0, 0.0, 0.0]], Round 1: [[0.0, 0.0, 0.0, 0.0, 0.0, 1.0], [0.0, 0.0, 0.0, 0.0, 0.0, 0.0], [0.0, 0.0, 0.0, 0.0, 0.0, 0.0], [0.0, 0.0, 0.0, 0.0, 0.0, 0.0], [0.0, 0.0, 0.0, 0.0, 0.0, 1.0], [1.0, 0.0, 0.0, 0.0, 1.0, 0.0]]]\n- Effort history of all agents: [Round 0: [2.86, 0.5, 4.28571429, 0.3, 1.0, 5.5], Round 1: [2.77, 1.0, 3.0, 0.3, 1.79, 4.6]]\n- Payoff history of all agents: [Round 0: [4.42827143, 0.125, 7.6477551, 0.045, 0.5, 11.37021429], Round 1: [2.91485, 0.0, 4.5, 0.045, 0.72305, 9.5664]]\n\nYou must detailedly explain your reasoning.\n\nAnd at the **end of your message**, Please return your final linking decision, which is a binary vector of length N = 6 **on a new line**, indicating whom you want to link with.\n- 1 means you want to link to that agent.\n- 0 means you do not.\n- You cannot link to yourself (position 0 must be 0).\n- Focus, Think and pay attention to which agent you are and your own parameters.\n\nOutput must be space-separated integers like: `0 1 0 1 0` (no explanation in that line).\n\nDo not swap the order. Always follow [Agent 0, Agent 1, Agent 2, ...] index order."}], "ability": "graph_decision", "reward_model": {"style": "custom", "ground_truth": [[0.0, 0.0, 1.0, 0.0, 0.0, 1.0], [0.0, 0.0, 0.0, 0.0, 0.0, 0.0], [1.0, 0.0, 0.0, 0.0, 1.0, 1.0], [0.0, 0.0, 0.0, 0.0, 0.0, 0.0], [0.0, 0.0, 1.0, 0.0, 0.0, 1.0], [1.0, 0.0, 1.0, 0.0, 1.0, 0.0]]}, "extra_info": {"round": 2, "index": 171, "groundtruth": [[0.0, 0.0, 1.0, 0.0, 0.0, 1.0], [0.0, 0.0, 0.0, 0.0, 0.0, 0.0], [1.0, 0.0, 0.0, 0.0, 1.0, 1.0], [0.0, 0.0, 0.0, 0.0, 0.0, 0.0], [0.0, 0.0, 1.0, 0.0, 0.0, 1.0], [1.0, 0.0, 1.0, 0.0, 1.0, 0.0]]}}
> ```

---

### Official Review · Reviewer_E7vm · 2025-10-30

**Soundness:** 4
**Presentation:** 3
**Contribution:** 3
**Rating:** 6
**Confidence:** 2

**Summary:**

This paper aims at improving LLMs' strategic decision making capability from a multi-agent perspective, which also consider decisions made by other agents. They first define a problem for real-world strategic decision-making and provide an environment as test bed. Then they evaluate the performance of current SOTA models with the environment, which also yields a dataset of strategic decisions made by the SOTA models, enabling further imitation learning for a smaller model. Then they propose a reinforcement learning algorithm called ToMPO (Theory of Mind Policy Optimization) to train a Qwen-2.5-7B-instruct. Results show that, with ToMPO, their trained agent is particularly good at achieving global welfare.

**Strengths:**

1. This paper addresses an important research question that how do we train an agent to make better decisions in a multi-agent environment. More specifically, better cooperative decision making. The paper tackles this problem from the perspective of considering strategies of others
2. Training with the method proposed by the paper achieves a good performance, even surpassing a much larger baseline.

**Weaknesses:**

1. Lack of case study. For example, presenting a decision-making trajectory could provide richer context and help readers better understand the problem.
2. Lack of in-depth analysis. For example, how are the hyper-parameters (w_{local} and w_{sample}) chosen? The reward design appears to be complex; providing further analysis could offer deeper insights into the design.
3. Lack of experiments. Applying the method to a single model seems to be insufficient. Does the current conclusion also hold for other models?

**Questions:**

1. Is the design specific to the two tasks evaluated in the paper, or can it be applied to a broader range of tasks? Providing an example that links it to other realistic cooperative multi-agent scenarios could help clarify the context.
2. Is it possible to include other MARL baselines, other than simple IO prompting?

---

> ### Author Response · Authors · 2025-11-25
> **Hyper-parameters and Analysis**
>
> Thanks for your professional and detailed opinions and your confirmation on our research motivation. We believe these questions help us enrich this paper.
>
> > Q1. Lack of in-depth analysis. For example, how are the hyper-parameters (w_{local} and w_{sample}) chosen? The reward design appears to be complex; providing further analysis could offer deeper insights into the design.
>
> This is indeed a crucial issue during training. Our experiments revealed that policy models achieve sample-level advantages more easily than graph-level advantages through rollout. This is because sample-level advantages are primarily numerical and confined to the same rollout group, while graph-level advantages involve comparisons with the historical best solutions of the same group and the network's hyper-parameters. For a model to gain an advantage at the graph level, it needs to learn graph-level information. This is also a difficult decision-making challenge when the model itself is an **actor** rather than an observer, making strategic decision-making a very important topic of discussion in multi-agent environments.
>
> Therefore, we aim for the model to dedicate most of its "energy" to learning at the sample level, and then use the remaining energy for graph-level learning, ultimately achieving a stable and appropriate advantage at both levels. This setup facilitates stable model training. We tested hyper-parameters of 4-6, but this proved challenging for the model to learn during the initial stages of training. Results showed significant reward collapse at steps 17, 42, and 47. However, the model trained more stably with hyper-parameters of 2-8. We observed that with settings 2-8, the model can complete graph-level learning, allowing us to maintain the hyper-parameter settings without modification.
>
> We will provide more analysis on model training, MAS, and decision-making in the updated version of the article.

---

> ### Author Response · Authors · 2025-11-25
> **Cases for Learn More about Decision-Making Process**
>
> > Q2. Lack of case study. For example, presenting a decision-making trajectory could provide richer context and help readers better understand the problem.
>
> We can learn the model's decision trajectory through two case studies.
>
> Case 1: By observing the behavior and payoffs of agents in previous rounds, Agent1 decides from its own perspective to continue cooperating with other individuals (Agent 2, 3, 4) who have higher effort, and not to cooperate with individuals (Agent 0) who have lower effort but higher payoff (who has the possibility to be a free-rider).
>
> ```
> Round 1 link matrix:
> [[0 0 1 1 1]
> [0 0 0 0 1]
> [1 0 0 0 0]
> [1 0 0 0 1]
> [1 1 0 1 0]]
> Round 1 efforts (e1): [1.  1.  1.  1.5 1.5]
> Round 1 efforts (e2): [0.8 0.8 1.5 2.  2. ]
> Round 1 payoffs: [3.14  2.52  0.232 0.352 4.66 ]
> Round 1 total welfare: 10.90
>
> Round 2: [Agent 1] Link decision reasoning:
> To make a strategic decision in this round, I will consider the following factors:
> 1. The cost of forming a link (c) is relatively high, so it's important to balance the benefits of connecting with another agent with the cost of doing so.
> 2. The efforts of other agents in the past rounds have been relatively high, so it's likely that they will also have high efforts in this round.
> 3. The group structure history shows that there have been links formed between agents in the past rounds, so it's reasonable to assume that agents will continue to connect with each other.
> Based on these factors, I will decide to form a mutual link with Agents 2, 3, and 4. This decision balances the cost of forming a link with the potential benefit of connecting with other agents who have shown high effort in the past rounds.
> Therefore, my final linking decision for this round is:
> `0 0 1 1 1`
> ```

---

> ### Author Response · Authors · 2025-11-25
>
> Case 2: As the number of rounds increases, the agent can utilize more historical information. The agent can then summarize the behavior of other agents in the environment from this historical information, thereby gradually understanding its own situation.
>
> Agent 0 Reasoning:
> ```
> Based on the historical information, it is evident that the agents have generally been choosing to link with a limited number of other agents, with Agent 0 and Agent 6 being the most commonly linked agents. The efforts exerted by the agents have also been decreasing over time, with Agent 0 expending the most effort in the initial rounds.
> Given that Agent 0 has a personal benefit parameter (α₀) of 3 and a cost per link (c) of 0.9, it is crucial for Agent 0 to balance the potential gains from linking with other agents with the costs of doing so. The global complementarity parameter (δ) is 0.12, which encourages cooperation between agents.
> To maximize Agent 0's payoff, it is essential to consider the efforts of other agents and the network structure. Since the efforts have been decreasing over time, it is possible that the agents have reached a point of diminishing returns, where further efforts would no longer significantly increase their payoffs. Therefore, it is advised for Agent 0 to focus on building and maintaining relationships with a few selected agents rather than trying to link with everyone.
> Based on the historical information, Agent 0 has been linked with Agent 3 in Round 0, Agent 2 in Round 1, Agent 3 in Round 2, Agent 5 in Round 6, and Agent 3 in Round 8. Agent 0 has not been linking with anyone else in the past rounds. Given this information, it appears that Agent 3 has been a relatively reliable partner for Agent 0, while the efforts of other agents have not justified the additional cost of linking with them.
> With this in mind, it is recommended for Agent 0 to continue linking with Agent 3 and not to link with any other agents. This decision is based on the following factors:
> 1. Agent 3 has been a reliable partner in the past, suggesting that there is a potential for a mutually beneficial relationship.
> 2. The efforts of other agents have been decreasing, indicating that they are not likely to significantly increase their payoffs with further efforts.
> 3. The cost per link is relatively high, and Agent 0 should be cautious about the number of links it forms.
> Therefore, Agent 0's decision on whom to link with is:
> `0 0 0 1 0 0 0`
> ```

---

> ### Author Response · Authors · 2025-11-25
>
> Case 3: After training with the GRPO algorithm, the model can also think about itself and other individuals in the environment. However, in terms of the thinking process, it is weaker than the results after training with the ToMPO algorithm. In terms of the results, the thinking process may not necessarily lead to appropriate decision-making results.
>
> Agent 0 Reasoning:
> ```
> To determine the best strategy for making a provisional linking decision, I will analyze the historical data and consider the potential impact of forming links on my payoff. Here's a detailed reasoning process:
>
> 1. **Effort History Analysis**: The effort history of all agents provides insight into their level of engagement. Agents who have been exerting more effort (higher values in the effort history) are likely to continue to do so in the next round, which could increase their payoff.
>
> 2. **Payoff Calculation**: The payoff formula includes the effort of the other agents, indicating that forming links with agents who have higher efforts can lead to a higher payoff.
>
> 3. **Linking Strategy**: Based on the effort history, I will focus on forming links with agents who have been exerting more effort and are likely to continue to do so in the next round. This will help me maximize my payoff.
>
> 4. **Risk Assessment**: I will also take into account the risk of forming links. Forming links with agents who have lower efforts or are less reliable could potentially decrease my payoff.
>
> Taking all of this into account, I will form links with agents 1, 2, 3, and 4, as they have been exerting more effort and are likely to continue to do so in the next round.
>
> Therefore, my provisional linking decision is to form links with agents 1, 2, 3, and 4.
>
> The binary vector indicating whom I want to link with is:
> `0 1 1 1 1`
> ```

---

> ### Author Response · Authors · 2025-11-25
> **Concerns about Baselines**
>
> > Q3. Is the design specific to the two tasks evaluated in the paper, or can it be applied to a broader range of tasks? Providing an example that links it to other realistic cooperative multi-agent scenarios could help clarify the context.
>
> > Q4. Is it possible to include other MARL baselines, other than simple IO prompting?
>
> We believe that real-world scenarios do indeed help to discover more of the model's capabilities, but it is necessary for us to elaborate on our problem definition and training methods.
>
> Unlike typical MAS works [1,2,3], our method decentralizes the training of individual models, allowing them to act as "participants" in choosing their own "position" for the next round, and to determine their own efforts based on the current graph. No single model participating in the game possesses an omniscient perspective or overall control. They are all individuals, relying on past outcome information to make decisions for the next round.
>
> Our method does not require all models to participate in training together; instead, only one model needs to be trained to improve decision-making capabilities from multiple agent perspectives.
>
> In the updated version of the paper, we will attempt to test real-world investment decisions beyond the cases mentioned above to refine our understanding of the model's capabilities.
>
>
>
> [1] Liao, J., Wen, M., Wang, J., & Zhang, W. (2025). Marft: Multi-agent reinforcement fine-tuning. *arXiv preprint arXiv:2504.16129*.
>
> [2] Liu, S., Chen, T., Liang, Z., Lyu, X., & Amato, C. (2025). Llm collaboration with multi-agent reinforcement learning. *arXiv preprint arXiv:2508.04652*.
>
> [3] Wan, Z., Li, Y., Wen, X., Song, Y., Wang, H., Yang, L., ... & Wen, Y. (2025). Rema: Learning to meta-think for llms with multi-agent reinforcement learning. *arXiv preprint arXiv:2503.09501*.

---

### Meta-Review · Area_Chair_MoLz · 2026-01-05

**Summary:**

The paper proposes ToMPO (Theory of Mind Policy Optimization), a reinforcement learning algorithm extending GRPO for training LLMs in strategic multi-agent decision-making. It formalizes sequential graph-effort decisions in cooperative games (BCZ variants and Public Goods Game extensions), using expert data from GPT-o3 for supervised fine-tuning and reinforcement against o3 opponents. ToMPO introduces dual-level (graph/sample) advantage estimation and balanced rewards to promote consideration of others' strategies.

Reviewers acknowledged the timely problem of multi-agent strategic reasoning in LLMs, the conceptual novelty of ToM-inspired credit assignment, and reported gains (e.g., outperforming larger models in compliance/cooperation). However, concerns focused on empirical clarity (ambiguous gains often limited to formatting/compliance rather than deeper reasoning), lack of ablations isolating components, heavy dependence on proprietary GPT-o3 (for both experts and opponents, risking imitation over independent strategy), limited baselines/MARL comparisons, unclear generalization beyond custom tasks, and presentation issues (dense notation, delayed definitions, inconsistent metrics).

The authors' rebuttal provided case studies, hyperparameter justifications, additional analyses (e.g., calibration, training curves), and clarifications on decentralization/ToM integration. These addressed some transparency issues but did not fully resolve questions on causal drivers of improvement, statistical robustness, or broader applicability.

**Reviewer Concerns:**

Clarifications on game settings, data examples, and hyperparameter choices were helpful. However, core issues remain outstanding: ablations for components, handling noisy/imperfect experts, statistical significance/variance, broader MARL baselines, and evidence of genuine ToM beyond action observation/imitation of o3.

**Reviewer Scores:**

Initial scores mixed (one 6, one 4, one 2). Rebuttal engagement yielded constructive dialogue but no clear upward revisions, with consensus leaning borderline-to-reject due to unresolved rigor and impact questions.

---

### Decision · Program_Chairs · 2026-01-26

Reject